# Women empowerment domains and unmet need for contraception among married and cohabiting fecund women in Sub-Saharan Africa: A multilevel analysis based on gender role framework

**Aklilu Habte**[1]*, **Aiggan Tamene**[1], **Biruk Bogale**[2]

**1** School of Public Health, College of Medicine and Health Sciences, Wachemo University, Hosanna, Ethiopia, **2** Department of Public Health, College of Medicine and Health Sciences, Mizan Tepi University, Mizan, Ethiopia

\* akliluhabte57@gmail.com

**Data Availability Statement:** The data supporting the findings of this study can be obtained in anonymized form from the Demographic and

## Abstract

### Background

Low women empowerment, is a known contributing factor to unmet needs for contraception by limiting access to health services through negative cultural beliefs and practices. However, little is known about the association between unmet needs and domains of women empowerment in Sub-Saharan African (SSA) countries. Hence, this study aimed at assessing the influence of women empowerment domains on the unmet need for contraception in the region using the most recent Demographic and Health Survey (DHS) data (2016–2021).

### Methods

The data for the study was derived from the appended women's (IR) file of eighteen SSA countries. A weighted sample of 128,939 married women was analyzed by STATA version 16. The Harvard Institute's Gender Roles Framework, which comprised of influencer, resource, and decision-making domains was employed to identify and categorize the covariates across three levels. The effects of each predictor on the unmet need for spacing and limiting were examined using a multivariable multilevel mixed-effect multinomial logistic regression analysis. Adjusted relative risk ratio (aRRR) with its corresponding 95% confidence interval was used to declare the statistical significance of the independent variables.

### Results

The pooled prevalence of unmet needs for contraception was 26.36% (95% CI: 24.83–30.40) in the region, with unmet needs for spacing and limiting being 16.74% (95% CI: 16.55, 17.02) and 9.62% (95% CI: 9.45, 12.78), respectively. Among variables in the influencer domain, educational level, family size of more than five, parity, number of children, attitude towards wife beating, and media exposure were substantially linked with an unmet

Health Survey website at https://www.dhsprogram.com upon reasonable request in the same manner as the authors. The authors did not have any special access privileges that others would not have.

**Funding:** The author(s) received no specific funding for this work.

**Competing interests:** The authors have declared that no competing interests exist.

**Abbreviations:** AIC, Akaike's information criterion; aRRR, Adjusted Relative Risk Ratio; CSA, Central Statistical Agency; DHS, Demographic and health survey; EA, Enumeration area; EDHS, Ethiopian Demographic and Health Survey; FP, Family Planning; GRF, Gender Roles Framework; GSEM, Generalized Structural Equation Modelling; ICC, Intra Class Correlation Coefficient; PCV, Proportional Change in Variance; RRR, Relative Risk Ratio; SSA, Sub Saharan Africa; VIF, Variance Inflation Factor.

need for spacing and limiting. Being in the poorest wealth quintile and enrollment in health insurance schemes, on the other hand, were the two variables in the resource domain that had a significant influence on unmet needs. The overall decision-making capacity of women was found to be the sole significant predictor of unmet needs among the covariates in the decision-making domain.

## Conclusion

Unmet needs for contraception in SSA countries were found to be high. Reproductive health program planners and contraceptive service providers should place due emphasis on women who lack formal education, are from low-income families, and have large family sizes. Governments should collaborate with insurance providers to increase health insurance coverage alongside incorporating family planning within the service package to minimize out-of-pocket costs. NGOs, government bodies, and program planners should collaborate across sectors to pool resources, advocate for policies, share best practices, and coordinate initiatives to maximize the capacity of women's decision-making autonomy.

## Introduction

In developing countries, more than 200 million women who wish to postpone or avoid pregnancy do not use family planning [1]. Rapid population growth in Sub-Saharan African (SSA) region is expected to continue until the end of the twenty-first century [2]. This region contains nearly all of the countries where fertility rates exceed five children per woman and where fertility rates are declining slowly [3, 4]. As per the Family Planning 2020 (FP2020) report, limited access to contraception services remains a major barrier to women's reproductive health in SSA regions [5]. Meanwhile, the proportion of unmet need for contraception among married women aged 15–49 in SSA is around 25% (i.e. about 47 million), compared to 14% in East Asia and 28% in Latin America and the Caribbean [6, 7]. This unmet need accounted for the majority (82%) of unintended pregnancies and related unsafe abortions and delayed or no prenatal care [8]. Unmet needs endanger women's health and quality of life by leading them directly to unwanted pregnancies, unsafe abortions, ill health, and related financial hardship for families and society [1].

Unmet need for contraceptive measures how far the nations have come toward achieving universal access to sexual and reproductive health services [9]. It is an indicator of the discrepancy between women's intentions for having children and their acceptance of contraception. Unmet needs for spacing and limiting are the two main categories. An unmet need for spacing is when a woman wants to delay or postpone getting pregnant, whereas an unmet need for limiting is when she wants to have no more children and is not using any form of contraception [10]. If all of the unmet needs for contraceptives are met, the number of unintended pregnancies in developing countries, including in the SSA sub-region, would drop from 75 million to 22 million annually [10]. With this, there would be 22 million, 15 million, and 90,000 fewer unintended pregnancies, unsafe abortions, and maternal deaths respectively [11, 12].

One vital dimension that has been shown to influence contraceptive use is women empowerment [13]. According to the World Bank, Women Empowerment(WE) is "the process of increasing a woman's or women's capacity to make meaningful choices and transform them into desired behaviors and results [14]. Others define it as having the ability to act,

independence, involvement, self-direction, consciousness, freedom, movement, and self-confidence [15, 16]. It has various indicators depending on various frameworks. The Harvard Analytical Framework (gender roles framework) developed by the Harvard Institute for International Development categorizes WE indicators into three dimensions: influencer, resource, and decision-making domains [17]. Influencing factors include variables that are symbolic of gender norms and beliefs, such as the gender division of labor, access to, and individual factors [18]. Resource factors include human capital and access to resources, whereas decision-making factors include women's participation in decision-making, including access to and use of resources [17, 18].

Women empowerment has the potential to increase contraceptive use [19, 20]. Low WE status, on the other hand, is a known contributing factor to unmet needs for contraceptives by limiting access to health service delivery points through negative cultural beliefs and practices [21, 22]. In some parts of the SSA, patriarchal rule impairs women's ability to exercise their fundamental reproductive health rights [23]. Some family planning initiatives in SSA failed to achieve their intended goal because they failed to take into account the power dynamics between women and their partners [24, 25]. On the other hand, studies have identified an association between unmet needs and indicators of WE like poor decision-making abilities, low media exposure, a lack of formal education, residence, parity, family size, access to maternity services, a high attitude towards wife beating, and unemployment [13, 26, 27].

Although the unmet need for contraceptive methods continues to be a major challenge in SSA countries, little is known about the relationship between unmet needs and domains of women empowerment in the region. In the realm of socio-economic progress and gender equality, studying the interplay between women's empowerment indicators and the pressing issue of unmet needs for contraception carries profound importance [28]. First of all, it paves the way for informed policies and actions to improve reproductive health outcomes, which have the potential to favorably impact the lives of millions of people. Furthermore, it will assist researchers and policymakers in gaining nuanced insights into the circumstances that hinder or facilitate women's power over reproductive decisions. Importantly, the inquiry has the potential to shed light on evidence-based policy initiatives that address both gender inequities and reproductive health gaps at the same time. Hence, using a gender analysis framework developed by the Harvard Institute, this study looked into the association between women's empowerment indicators and unmet contraceptive needs in 18 SSA countries. The study's findings will help to intervene in empowering women, which is critical to improving maternal and child health outcomes, lowering fertility rates, and reducing population growth—all of which have far-reaching implications for sustainable development, poverty reduction, and overall societal well-being.

## Methods

### Data source, population, and study period

A total of 18 demographic and health surveys (DHS) carried out in SSA countries between 2016 and 2021 were considered in this study. The DHS is conducted in 90 countries to collect information on fundamental health indicators. The data for this study came from the appended women's (IR) file, which contains information about contraception, and all important covariates. The study included countries with recent standardized DHS reports from 2016 to 2020 that had complete cases on the relevant variables. Only married and cohabited women who were exposed to frequent sexual activity were included. Whereas, infecund women who were unable to conceive owing to biological issues were not supposed to utilize contraception

**Table 1. Description of the SSA countries included in the analysis with their respective sample size, 2016–2021.**

| Countries | DHS Year | Weighted sample size [n(%)] |
|---|---|---|
| Angola | 2016 | 1,229(0.95) |
| Cameroon | 2018 | 4,788(3.71) |
| Burundi | 2017 | 6,450(5.00) |
| Ethiopia | 2016 | 8,844(6.86) |
| Madagascar | 2021 | 8,820(6.84) |
| Malawi | 2017 | 14,190(11.01) |
| Mauritania | 2020 | 8,236(6.39) |
| Rwanda | 2020 | 4,277(3.32) |
| Uganda | 2016 | 5,045(3.91) |
| Zambia | 2018 | 6,878(5.33) |
| Benin | 2018 | 7,435(5.77) |
| Gambia | 2020 | 6,348(4.92) |
| Guinea | 2018 | 5,662(4.39) |
| Liberia | 2021 | 1,586(1.23) |
| Mali | 2018 | 7,142(5.54) |
| Nigeria | 2018 | 23,408(18.15) |
| Sierra Leone | 2020 | 7,010(5.44) |
| South Africa | 2016 | 1,591(1.23) |
| **Total** | **2016–2021** | 128,939(100.00) |

and were therefore excluded [29]. The entire analysis relied on a weighted sample of 128,939 married and fecund women (Table 1).

## Data collection tool and procedures

Using structured questionnaires developed in each country's official language, the data were collected through face-to-face interviews with well-trained data collectors. The respondents for the survey were selected using a two-stage cluster sampling method. The first stage involved the selection of enumeration areas (EAs) and household listing within the selected EAs based on the most recent population and housing census data from each country as a sampling frame. Households were chosen in the second stage using an equal probability sampling criterion. The Demographic and Health Survey Sampling and Household Listing Manual developed by ICF International goes into great detail about the sampling method used during the DHS [30].

## Measurement of variables of the study

The outcome variable was unmet need for contraception. The interviewers asked married women to describe their pattern of using contraceptive methods, and their response was labeled under one of the following categories: *Has unmet need for spacing*; *Has unmet need* for *limiting*; *Using for spacing*; *Using for limiting*; *No unmet need*; *Infecund and menopausal*. Menopausal or infecund women were excluded from the dataset before analysis because they had no demand for contraceptive uptake [29]. Finally, three categories were formed based on the DHS's recently revised definition of unmet need for contraception:

i. Unmet needs for spacing (women who wanted to delay having another child but were not using any form of contraception)

ii. Unmet needs for limiting (women who did not want any more children, but did not use any contraceptive method), and

iii. No unmet need

**Women's empowerment variables.**   To select and categorize variables across different levels, the study used the Gender Roles Framework (GRF) developed by the Harvard Institute for International Development [17, 18, 31]. The framework categorizes women empowerment variables into; influencer, resource, and decision-making factors [17] (Table 2).

## Data management and statistical analyses

By using STATA/SE version 16.0, DHS reports of 18 SSA countries were appended, recoded, cleaned, and analysed. Weighting was applied to the data to restore the survey's statistical representativeness and to get an appropriate statistical estimate. In order to account for the uneven likelihood of selection between the strata caused by the non-proportional distribution of samples to different sub-regions, residences, and non-response rates, a weighted analysis was carried out. Descriptive statistics such as frequencies, percentages, and means were computed to describe the study participants' background characteristics. The number and proportion of women with all three types of unmet needs were presented (total unmet need, unmet need for spacing, and unmet need for limiting). Chi-square tests were used to examine the distribution of covariates across various categories of unmet needs. The Variance Inflation Factor (VIF), which was employed to check for the presence of multicollinearity between the variables, revealed that there was none (the VIF ranges from 1.01 to 2.98 with a mean of 1.49).

**Multilevel multinomial regression.**   Multilevel modeling was used due to the hierarchical nature of the DHS data, where women were nested within families and households were nested within clusters. Using a multilevel analysis for such hierarchical data allows us to avoid biased parameter and standard error estimations that could occur with a single-level study [36]. A multilevel mixed-effect multinomial logistic regression was a good fit for the current study because the outcome variable contained more than two categories (no unmet need, unmet need for spacing, and unmet need for limiting).

**Modelling approaches.**   For the overall analysis, three modeling approaches were used: fixed effects, random effects, and goodness of fit.

**Fixed effect model.**   First, a bivariable multilevel multinomial regression was fitted to select eligible variables for the multilevel multivariable regression, and variables with a p-value less than 0.20 were eligible for a multilevel mixed-effect multivariable multinomial logistic regression. To estimate the model, we used generalized structural equation modeling (GSEM) with "gsem" command in STATA. The GSEM model evaluated both the fixed effects of various explanatory variables and the random effects at the cluster level. After the analysis has been completed, the adjusted relative risk ratio (aRRR) and its 95% confidence interval (CI) were reported, and variables with a p-value less than 0.05 were deemed to be significant predictors of unmet family planning (unmet need for spacing and limiting)

**Random effect modelling.**   By employing a hierarchical approach, five distinct nested models were fitted: model 1 (a null model (without any covariates), Model 2 (influencer domain), Model 3 (a resource domain), model 4 (a decision-making domain), and Model 5 (the combination of all three domains).

The presence of variability of unmet need for contraception across clusters/communities was evaluated using proportional change in variance (PCV) and the intraclass correlation coefficient (ICC).

**Table 2. List of covariates in the influencer, resource, and decision-making domains that affect unmet need for contraception in SSA countries 2016–2021.**

| Influencer domain(Individual-level factors) | |
|---|---|
| Variables | Description |
| Maternal age | 15–24, 25–34, 35 and above |
| Residence | Urban, rural* |
| Region | Central region: Angola, Cameroon, <br> Eastern region: Burundi, Ethiopia, Madagascar, Malawi, Rwanda, Uganda, Zambia, and Mauritania <br> Western region: Benin, Gambia, Guinea, Liberia, Mali, Nigeria, and Sierra Leone <br> Southern*: South Africa |
| Family size | ≤5memeber, >5 member* |
| Sex of head of Household | Female, Male* |
| Parity | Nulliparous*, primiparous, multiparous, Grand multiparous |
| Total children ever born | 0*, 1, 2–4 and ≥5 |
| Pregnancy status during last childbirth | Unintended, Intended* |
| Knowledge of Contraceptive methods | No, Yes* |
| Exposure to media (reading a newspaper, listening to the radio, and watching television) | Not at all, Less than once a week, At least once a week* |
| Women's and husbands' education | No education, primary, secondary, or higher education* |
| Getting permission to go to a health facility for medical care | Not a big problem, a big problem* |
| Community Education | The proportion of respondents in the cluster who reported having completed primary, secondary, or higher education. The cluster's overall educational accomplishment may be determined by adding the primary, secondary, and higher education levels of each respondent and categorized as: <br> Low, and high* |
| The overall attitude towards Wife beating [a] | Low, moderate, and High* |
| **Resource domain** | |
| Occupation | Employed* and unemployed |
| Wealth index | Richest*, richer, middle, poorer, poorest |
| Having a mobile phone | Yes, no* |
| Getting money needed for treatment | Not a big problem, a big problem* |
| Distance to a health facility | Not a big problem, a big problem* |
| Being a member of health insurance schemes | Yes, no* |
| Community level poverty | Described as the proportion of respondents who resided in the poorest housing stock in the cluster. By adding up the individual households with the lowest wealth indices, the cluster's overall poverty can be calculated, and categorized as Low, moderate, high* |
| **Decision making domain** | |
| A decision in accessing healthcare | By woman alone, by a woman and her partner, by a partner and other third parties* |
| Decision on major purchases | |
| Decide to visit family | |
| Overall decision-making [b] | |
| | *Reference for the categories |

[a] **Attitude toward wife beating:** Acceptance of wife beating was measured by using five items: *(i)* Beating justified if she neglects children, *(ii)* Beating justified if she argues with her husband, *(iii)* Beating justified if she refuses to have sex, *(iv)* beating justified if she goes out without the permission of her husband, and *(v)* beating justified if she burns foods. The response categories for each item were *(i)* no, *(ii)* yes and *(iii)* don't know. Response (i) was given a value of 0, indicating that it was not accepted, and the other responses were given values of 1, indicating that they agreed on wife beating. Responses to those five items were combined to produce a composite score, which ranges from 0 to 5. A lower score on this indicator is interpreted as reflecting a greater sense of entitlement and self-esteem and a higher status of women [32]. Finally, a composite score was grouped into three categories; low, middle, and high for scores "0 to 2", "3 to 4", and "5" respectively [33, 34].

[b] **Overall decision-making power:** Was evaluated using answers to questions about who makes final decisions for the family when it comes to major purchases for the home, visits to family, and health care. (i) respondent alone, (ii) respondent and husband/partner, (iii) husband/partner alone, (iv) someone else, and (v) others were the response categories. For each question, (i) or (ii) responses were given a value of 1, indicating high decision-making power and the other responses were given a value of 0, denoting low power. Responses to the three decision-making power dimensions were combined to produce a composite score, which ranges from 0 to 3. This index is positively related to women's empowerment and reflects the level of decision-making power that women have in areas affecting their own lives and environments. Finally, a composite score was grouped into three categories; low, middle, and high for scores "0 to 1", "2", and "3" respectively [34, 35].

$ICC = \frac{var(b)}{Var(b)+Var(w)}$, where Var(b) is the variance at the group level and Var(w) is a predicted individual variance component, which is $\pi^2/3 \approx 3.29$

Proportional Change in Variance (PCV) was estimated as

$PCV = \frac{(Va-Vb)}{Va}*100$, where, $V_a$ is the variance of the initial model (null model), and $V_b$ = variance of the subsequent models (models 2, 3, 4, and 5).

**Goodness of fit.** The goodness of fit was also assessed using deviance = (-2 * (Log Likelihood (LL), Schwarz's Bayesian Information Criteria (BIC), and Akaike's Information Criterion (AIC). Lower AIC or BIC values are regarded as better explanatory models. After a comparison of all models, the fifth model with the lowest deviance, AIC and BIC was chosen as the best-fit model (Table 6).

## Ethics approval and consent to participate

All practices and procedures followed the appropriate laws and regulations outlined in the Helsinki Declaration. The DHS survey report did not include an ethical approval ID, but the authors received written permission from ICF International (DHS office) to access this dataset upon registration with possible justification. The data, which were only used for authorized study, were accessible to only the authors. The data were kept private, and no attempt was made to identify any family or individual. The DHS added that at the time of the initial data collection, every person and/or their legal guardians gave their written consent. Additionally, the authors obtained ethical approval from the Institutional Review Board (IRB) of Wachemo University College of Medicine and Health Sciences to guarantee that the research procedure was carried out in a manner that complied with ethics both domestically and internationally.

## Results

### The distribution of characteristics of the respondents under the 'influencer' domain

This study included a weighted sample of 128,939 married women of reproductive age from 18 SSA countries. The Eastern and Western sub regions, respectively, contributed 48.7% and 45.4% of all respondents. Most, 23,408 (18.15%) of the study participants were from Nigeria (**Table 1**). The average age of the study participants was 30.38 (SD±7.85) years, with the majority (61.1%) falling between the ages group 20–34 years. Almost two-fifths (39.4%) and more than two-thirds (67.8%) of study participants had no formal education and lived in rural areas, respectively. Of the respondents, 86.7%, 62.1%, and 45.4%, never had access to a newspaper, television, or radio, respectively. Grand multiparous respondents (women with five or more living children) made up the second-largest group of respondents (27.7%), following multiparous respondents (49.8%). The majority of women (86.5%) were from male-headed households, and 54.1% of them were from a family size of five or more people. Modern family planning methods have been known to 96.4% of the respondents. Women aged 20–34 had the largest proportion of unmet need for spacing contraceptives (69.8%), followed by rural residents (67.7%) and multiparous (54.1%) (p-value <0.001) (Table 3). Nearly three-quarters (73.4%) of women had a low attitude towards wife beating. When compared to women who have no unmet need, the proportion of women who accept wife beating in multiple circumstances is higher among women who have an unmet need for spacing and limiting (Fig 1).

**Table 3. Distribution of factors under the 'influencer' domain across different categories of unmet need for contraception in SSA, 2016–2021.**

| Variable categories | Weighted sample size (N = 128,939) [n(%)] | Unmet need for spacing (N = 21,582) [n(%)] | Unmet need for limiting (N = 12,401) n (%) | No Unmet need (N = 94,955) [n(%)] | Test statistics $X^2$ |
|---|---|---|---|---|---|
| **Sub-Regions** | | | | | |
| Eastern | 62,738(48.7) | 9,162(42.4) | 6,276(50.6) | 47,300(49.8) | 698.55** |
| Western | 58,592(45.4) | 11,137(51.6) | 5,261(42.4) | 42,195(44.4) | |
| Central | 6,018(4.7) | 1,185(5.5) | 675(5.5) | 4,158(4.4) | |
| Southern | 1,591(1.2) | 98(0.5) | 190(1.5) | 1,303(1.4) | |
| **Current Age** | | | | | |
| 15–19 | 9,322(7.2) | 1,860(8.6) | 87(0.7) | 7,374(7.7) | 1313.2** |
| 20–34 | 78,839(61.1) | 15,062(69.8) | 3,572(28.8) | 60,205(63.4) | |
| 35–49 | 40,778(31.6) | 4,661(21.6) | 8,742(70.5) | 27,375(28.8) | |
| Age at cohabitation | | | | | |
| <18 years | 60,675(47.1) | 10,4689(48.5) | 5,935(47.9) | 44,272(46.6) | 9.57 |
| ≥18 years | 68,264(52.9) | 11,113(51.5) | 6,467(52.1) | 50,683(53.4) | |
| **Educational status** | | | | | |
| No education | 50,818(39.4) | 9,421(43.7) | 5,797(46.8) | 35,599(33.5) | 786.15** |
| Primary | 42,059(32.6) | 6,654(30.8) | 4,097(38.0) | 31,308(33.0) | |
| Secondary | 29,561(22.9) | 4,699(21.8) | 2,087(16.8) | 22,774(24.0) | |
| Higher | 6,502(5.0) | 808(3.7) | 420(3.4) | 5,273(5.5) | |
| **Residence** | | | | | |
| Urban | 41,513(32.2) | 7,050(32.7) | 3,891(31.4) | 30,572(32.2) | 2.10 |
| Rural | 87,426(67.8) | 14,532(67.3) | 8,511(68.6) | 64,383(67.7) | |
| **Family size** | | | | | |
| ≤5memeber | 59,155(45.9) | 9,229(42.8) | 3,376(27.2) | 46,550(49.0) | 2001.10** |
| >5 member | 69,784 (54.1) | 12,353(57.2) | 9,026(72.8) | 48,405(51.0) | |
| **Head of household** | | | | | |
| Male | 111,581(86.5) | 17,859(82.8) | 10,479(84.5) | 83,243(87.7) | 321.73** |
| Female | 17,358(13.5) | 3,723(17.2) | 1,922(15.5) | 11,712(12.3) | |
| **Community Education** | | | | | |
| Low | 45,504(35.3) | 7,432(34.5) | 4,487(36.2) | 33,585(33.4) | 30.81* |
| Moderate | 42,683(33.1) | 7,347(34.0) | 4,134(33.3) | 31,202(32.9) | |
| High | 40,751(31.6) | 6,802(31.5) | 3,781(30.5) | 30,168(31.7) | |
| **Parity** | | | | | |
| Nulliparous | 8,721(6.8) | 931(4.3) | 49(0.4) | 7,739(8.1) | 2118.31** |
| Primiparous | 20,312(15.8) | 3,8456(17.8) | 287(2.3) | 16,180(17.0) | |
| Multiparous | 64,193(49.8) | 11,673(54.1) | 4,208(33.9) | 48,312 (50.9) | |
| Grand multiparous | 35,712 (27.7) | 5,131(23.8) | 7,857(63.4) | 22,724(24.0) | |
| **Children ever born** | | | | | |
| No children | 7,822(6.1) | 856(4.0) | 42(0.3) | 6,923(7.3) | 1918.22** |
| 1 | 18,499(14.3) | 3,499(16.2) | 247(2.0) | 14,753(15.5) | |
| 2–4 | 59,507 (46.1) | 10,830(50.2) | 3,423(27.6) | 45,254(47.7) | |
| ≥5 | 43,112 (33.4) | 6,397(29.6) | 8,690(70.1) | 28,025(29.5) | |
| Knowledge of modern contraceptive | | | | | |
| Yes | 124,572(96.6) | 32,768(96.4) | 12,035(97.0) | 91,804(96.7) | 29.42* |
| No | 4,367(3.4) | 1,216(3.6) | 366(3.0) | 3,151(3.3) | |
| **Reading newspaper** | | | | | |
| Not at all | 111,820(86.7) | 19,468(90.2) | 10,955(88.3) | 81,397(85.7) | 322.98** |

(*Continued*)

**Table 3.** (Continued)

| Variable categories | Weighted sample size (N = 128,939) [n(%)] | Unmet need for spacing (N = 21,582) [n(%)] | Unmet need for limiting (N = 12,401) n (%) | No Unmet need (N = 94,955) [n(%)] | Test statistics X² |
|---|---|---|---|---|---|
| Less than once a week | 10,676(8.3) | 1,386(6.4) | 885(7.2) | 8,405(8.9) | |
| At least once a week | 6,530 (5.0) | 728(3.4) | 551(4.5) | 5,152.5(5.4) | |
| **Listening to a radio** | | | | | |
| Not at all | 58,597(45.4) | 10,457(48.4) | 5,739(46.3) | 42,401(44.7) | 172.57** |
| Less than once a week | 28,087(21.8) | 4,876(22.6) | 2,536(20.5) | 20,674(21.8) | |
| One and more times a week | 42,255(32.8) | 6,249(29.0) | 4,127(33.3) | 31,879(33.5) | |
| **Watching TV** | | | | | |
| Not at all | 80,040(62.1) | 13,417(62.2) | 7,984(64.4) | 58,639(61.8) | 84.02** |
| Less than once a week | 17,909(13.9) | 3,289(15.2) | 1,619(13.0) | 13,002(13.7) | |
| One and more times a week | 30,989 (24.0) | 4,876(22.6) | 2,799(22.6) | 23,314(24.5) | |
| **Overall Acceptance of wife beating** | | | | | |
| Low | 94,640 (73.4) | 15,482(71.7) | 9,245(74.5) | 69,913(73.6) | 93.81** |
| Medium | 20,870(16.1) | 3,747(17.4) | 1,925(15.5) | 15,198(16.0) | |
| High | 13,428(10.4) | 2,353(10.9) | 1,232(10.0) | 9,843(10.4) | |

** p-values<0.001

* p-values<0.05

## The distribution of characteristics of the respondent under the 'resource' domain

Three quarter (75.0%) of women were employed. A lack of money and a long distance have been mentioned as major barriers to obtaining healthcare facilities by 49.0% and 37.4% of

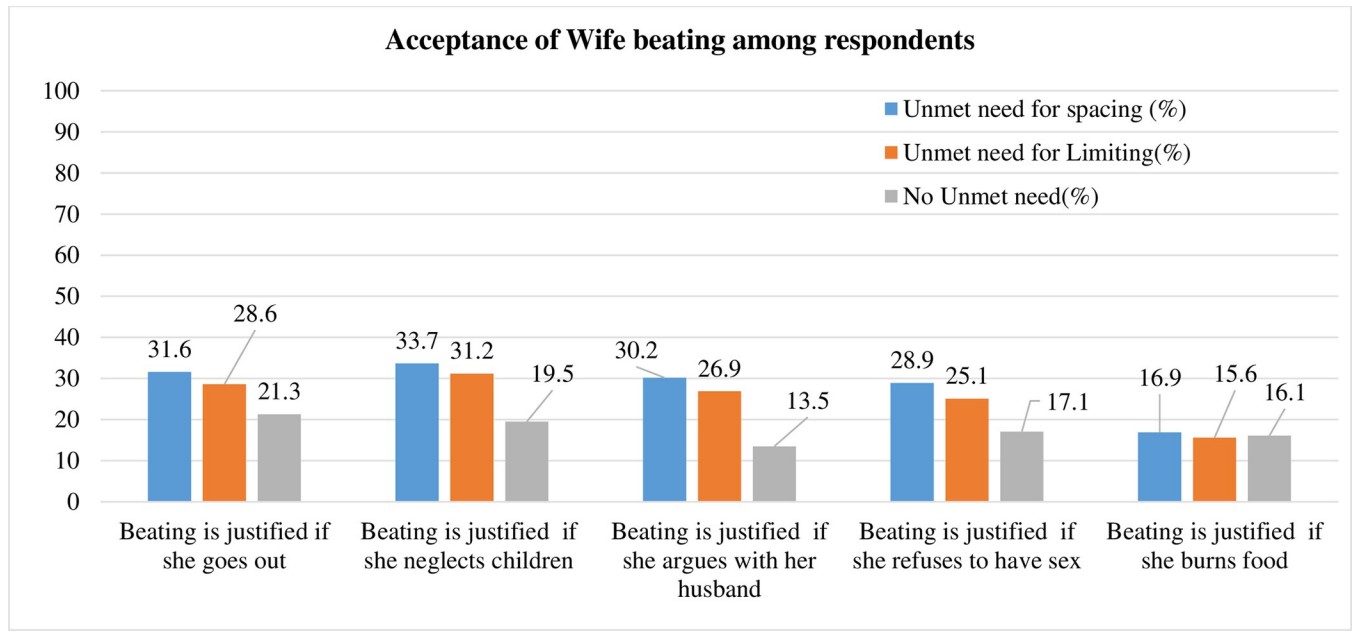

**Fig 1. The proportion of women in SSA countries who positively accept wife beating, 2016–2021.**

**Table 4. Distribution of factors under resource domain across different categories of unmet need for contraception in SSA, 2016–2021.**

| Variable categories | Weighted sample size (N = 128,939) [n (%)] | Unmet need for spacing (N = 21,582) [n (%)] | Unmet need for limiting (N = 12,401) n (%) | No Unmet need (N = 94,955) [n(%)] | Test statistics X² |
|---|---|---|---|---|---|
| **Wealth index combined** | | | | | |
| Poorest | 25,650(19.9) | 4,742(22.0) | 2,453(19.8) | 18,456(19.4) | 301.72** |
| Poorer | 26,514(20.6) | 4,713(21.8) | 2,561(20.7) | 19,239(20.3) | |
| Middle | 25,602(19.9) | 4,421(20.5) | 2,544(20.5) | 18,636(19.6) | |
| Richer | 25,304 (19.6) | 4,051(18.8) | 2,446(19.7) | 18,806(19.8) | |
| Richest | 25,869(20.0) | 3,655(16.9) | 2,397(19.3) | 19,818(20.9) | |
| Community poverty | | | | | |
| Low | 44,908(34.8) | 7,425(34.4) | 4,352(35.1) | 33,130(34.9) | 26.02* |
| Moderate | 42,428(32.9) | 7,056(32.7) | 4,161(33.5) | 31,211(32.9) | |
| High | 41,602(32.3) | 7,100(32.9) | 3,888(31.3) | 30,614(32.2) | |
| **Distance to a health facility** | | | | | |
| Big problem | 48,291(37.4) | 8,684(40.2) | 4,853(39.1) | 34,754(36.6) | 89.23** |
| Not a big problem | 80,648(62.6) | 12,899(59.8) | 7,548(60.9) | 60,201(63.4) | |
| **Money needed for treatment** | | | | | |
| Big problem | 63,196 (49.0) | 11,318(52.4) | 6,682(53.9) | 45,196(47.6) | 280.84** |
| Not a big problem | 65,743 (51.0) | 10,264(47.6) | 5,720(46.1) | 49,759(52.4) | |
| **Covered by Health insurance** | | | | | |
| Yes | 9,896 (7.7) | 1,177(5.5) | 983(7.9) | 7,736(8.2) | 206.08** |
| No | 118,224(92.3) | 20,405(94.5) | 11,419(92.1) | 87,219(91.8) | |
| **Occupational status** | | | | | |
| Employed | 96,661(75.0) | 15,006(69.5) | 9,772(78.8) | 71,884(75.7) | 414.67** |
| Un employed | 32,278(25.0) | 6,577(30.5) | 2,630(21.2) | 23,071(24.3) | |

\*\* p-values<0.001

\* p-values<0.05

women, respectively. Unmet needs for both spacing and limiting were higher among employed women, at 69.5% and 78.8% respectively (p-value <0.001) (Table 4).

## The distribution of characteristics of women under the decision-making domain

In all, 38.8% of women lacked autonomy. When it came to large household purchases, accessing healthcare, and visiting relatives and friends, respectively, 44.2%, 41.0%, and 32.4% of women did not participate in decision-making. The unmet need for spacing is higher among those non-autonomous groups (p-value <0.001) (Table 5).

## The pooled prevalence of unmet need for contraception in SSA countries

For all eighteen SSA countries in the four sub-regions, the pooled prevalence of unmet need for contraception was 26.36% (95% CI: 24.83–30.40). Unmet need for spacing and limiting was 16.74% (95% CI; 16.55, 17.02) and 9.62% (95% CI; 9.45, 12.78), respectively. The prevalence varied by country, ranging from 15.8% (95% CI: 115.0–16.50%) in Madagascar to 42.0% (95% CI: 39.30–44.8%) in Angola. When it came to geographical disparities, the central and western areas had the highest burden of unmet needs, accounting for 34.94% (95% CI: 21.22–48.66) and 31.13% (95% CI: 26.53–35.72), respectively (Fig 2).

**Table 5. Distribution of factors under decision making domain across different categories of unmet needs for contraception in SSA countries, 2016–2021.**

| Variable categories | Weighted sample size (N = 128,939) [n (%)] | Unmet need for spacing (N = 21,582) [n (%)] | Unmet need for limiting (N = 12,401) [n (%)] | No Unmet need (N = 94,955) [n(%)] | Significance (X²) |
|---|---|---|---|---|---|
| **Decision maker for healthcare** | | | | | |
| Woman alone | 21,210(16.5) | 3,156(14.6) | 2,640(21.3) | 15,414(16.2) | 533.03** |
| Woman and partner | 54,834(42.5) | 8,336(38.6) | 5,443(43.9) | 41,055(43.2) | |
| Husband and others | 52,896(41.0) | 10,090(46.8) | 4,320(34.8) | 38,486(40.5) | |
| **Decision maker for major purchases** | | | | | |
| Woman alone | 12,757(9.9) | 2,154(10.0) | 1,833(14.8) | 8,771(9.2) | 730.64** |
| Woman and partner | 59,152(45.9) | 8,700(40.3) | 5,891(47.5) | 44,560(46.9) | |
| Husband and others | 57,025(44.2) | 10,727(49.7) | 4,678(37.7) | 41,624(43.9) | |
| **Decide to visit family** | | | | | |
| Woman alone | 21,261(16.5) | 3,485(16.2) | 2,579(20.8) | 15,197(16.0) | 457.95** |
| Woman and partner | 65,903(51.1) | 10,059(46.6) | 6,329(51.0) | 49,514(52.1) | |
| Husband and others | 41,770(32.4) | 8,038(37.2) | 3,493(28.2) | 30,244(31.9) | |
| **Overall decision-making power** | | | | | |
| Low | 50,085(38.8) | 9,659(44.8) | 4,035(32.5) | 36,390(38.3) | 513.45** |
| Medium | 19,455(15.1) | 3,196(14.8) | 1,875(15.1) | 14,384(15.2) | |
| High | 59,400(46.1) | 8,727(40.4) | 6,492(52.4) | 44,181(46.5) | |
| **Permission to go to a health facility** | | | | | |
| Big problem | 22,228(17.2) | 4,486(20.8) | 2,263(18.3) | 15,478(16.3) | 212.79** |
| Not a big problem | 106,711(82.8) | 17,096(79.2) | 10,138(81.7) | 79,476(83.7) | |

** P-value <0.001

* p-values<0.05

### Random effect (measures of variation)

The Intraclass Correlation Coefficient (ICC) and Proportional Change in Variance (PCV) statistics have been computed for the measures of variation (random effects). The results of the empty model (model 1), revealed that variations between clusters account for 13.8% of the total variance in unmet needs (ICC = 0.138, p<0.001). Variables in the influencer, resource, and decision-making domains collectively accounted for 64.1% of the variation seen in the null model (PCV = 64.1%). The values of AIC, BIC, and Deviance decreased continually as we moved from model 1 (the empty model) to model 5 (the full model), indicating that the final model created throughout the study had satisfactory goodness of fit. Finally, the fifth model with the lowest deviance (176146.2) was selected as the best-fit model after comparison (Table 6).

**Fixed effects: Factors associated with unmet need for contraception.** *A bivariable multilevel multinomial regression analysis.* Factors in the influencer domain like region, age of respondents, educational status, family size, parity, knowledge of contraceptive methods, listening to the radio, and reading newspapers were associated with both unmet need for spacing and limiting (Table 7).

Factors in the resource domain namely wealth index, enrolment in the health insurance scheme, employment status, distance to a health facility, and shortage of money were associated with unmet need for spacing and limiting(Table 8).

Similarly, the bivariable multilevel multinomial logistic regression discovered a link between the unmet need for both spacing and limiting and respondents' decision-making capacity in healthcare, big purchases, and visits to family or friends (Table 9).

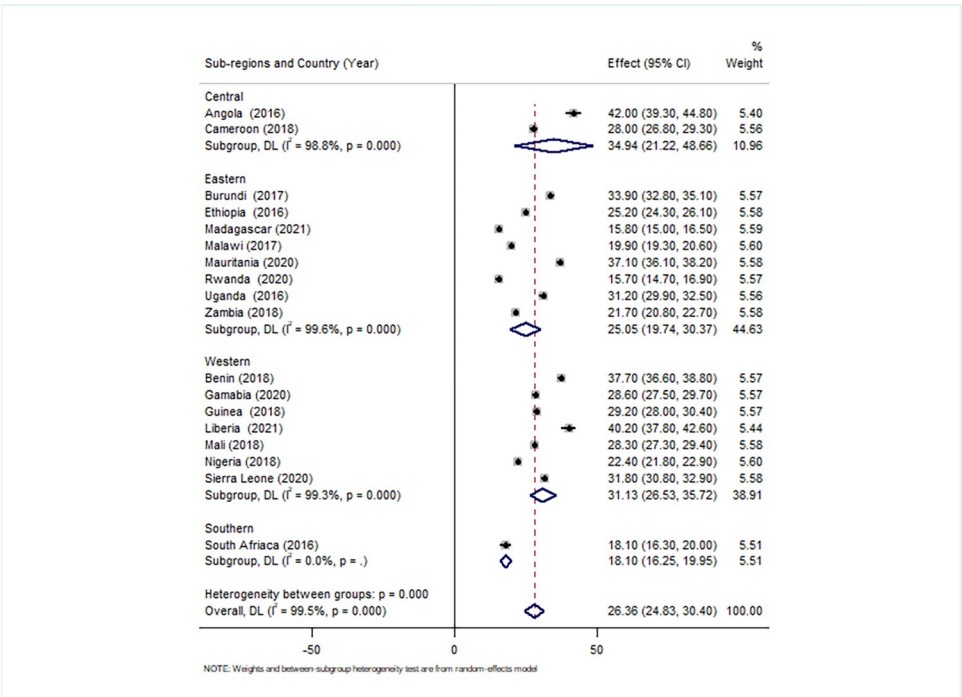

**Fig 2. A forest plot depicts the country and regional prevalence of overall unmet needs for contraception in SSA countries, 2016–2021.**

*A multilevel-mixed effect multivariable multinomial logistic regression.* To identify the factors associated with the unmet need for contraception, a multilevel, multivariable, multinomial logistic regression analysis was performed. Variables within the influencer, resource, and choice domains were examined in this study's regression analysis. Among the influencer domains, lack of formal education, family size of >5, parity, the total number of children, attitude toward wife beating, and listening to the radio were significantly associated with an unmet need for spacing and limiting. Among variables in the resource domain, being in the poorest wealth quintile, and not enrolled in HI schemes were identified as significant determinants of unmet need for spacing and limiting. Low decision-making power was the only covariate under the decision domain that influence both unmet needs.

Women with no formal education were more likely to experience unmet needs for spacing and limiting, respectively, by 13% (aRRR = 1.13; 95% CI: 1.03; 1.24) and 34% (aRRR = 1.34;

**Table 6. Random intercept variances and model fit statistics comparison of multilevel mixed effect multinomial logistic regression model.**

| Measures | Model 1 (null model) | Model 2 (influencer domain) | Model-3 (resource domain) | Model 4 (decision-making domain) | Model 5 Full model |
|---|---|---|---|---|---|
| **Random effects** | | | | | |
| Variance | 0.53 | 0.41 | 0.36 | 0.23 | 0.19 |
| ICC | 13.8% | 11.1% | 9.9% | 6.5% | 5.5 |
| AIC | 192161.1 | 187028.5 | 191139.8 | 181130.8 | 176336.1 |
| BIC | 192190.4 | 187584.7 | 191364.2 | 181335.7 | 177263.0 |
| PCV | Reference | 22.64% | 32.02% | 56.6% | 64.1 |
| **Model fitness** | | | | | |
| Log-likelihood | -96077.6 | -88457.2 | -95546.9 | -96,304.04 | -88073.1 |
| Deviance | 192155.2 | 176914.4 | 176914.4 | 192608 | 176146.2 |

**Table 7. Multilevel bivariable multinomial logistic regression analysis of factors in influencer domains affecting unmet need for contraception in SSA countries, 2016–2021.**

| Variable categories | Unmet need for spacing cRRR (95% CI) | Unmet need for limiting cRRR (95% CI) |
|---|---|---|
| **Sub-Regions** | | |
| Southern | 1 | |
| Central | 3.79(2.84, 5.06)* | 1.14(0.89, 1.38)* |
| Eastern | 2.57 (1.95, 3.40)* | 0.81(0.75, 1.11) |
| Western | 3.51(2.66, 4.64)* | 0.85(0.70, 1.04) |
| **Current Age** | | |
| 15–19 | 1 | 1 |
| 20–34 | 0.99(0.93, 1.05)* | 5.00(3.78, 6.61)* |
| 35–49 | 0.67(0 .63, 0.72*) | 13.2(10.39, 15.5)* |
| **Educational status** | | |
| Higher | 1 | |
| Secondary | 1.34(1.21, 1.49)* | 1.15(0.99, 1.32)* |
| Primary | 1.39 (1.25, 1.53)* | 1.64(1.43, 1.88)* |
| No education | 1.73(1.56, 1.90) * | 2.04(1.79, 2.33)* |
| **Family size** | | |
| ≤5memeber | 1 | 1 |
| >5 member | 1.29(1.24, 1.33)* | 2.57(2.44, 2.70)* |
| **Head of household** | | |
| Female | 1 | 1 |
| Male | 0.67(0.64, 0.70)* | 0.76(0.72, 0.82)* |
| **Community Education** | | |
| High | 1 | 1 |
| Moderate | 1.04(1.01, 1.09) | 1.05(0.99, 1.11) |
| Low | 0.98(0.94, 1.12) | 1.06(1.01, 1.13) |
| **Parity** | | |
| Nulliparous | 1 | 1 |
| Primiparous | 1.97(1.79, 2.17)* | 2.77(1.86, 4.13)* |
| Multiparous | 2.00(1.84, 2.19)* | 13.63(9.47, 19.62)* |
| Grand multiparous | 1.87(1.71, 2.05)* | 24.12(17.63, 37.84)* |
| **Children ever born** | | |
| No children | 1 | 1 |
| One | 1.92(1.74, 2.11)* | 2.74(1.76, 4.27)* |
| Two to four | 1.93(1.76, 2.12)* | 12.80(8.28, 18.58)* |
| Five and more | 1.84(1.68, .03)* | 20.86(13.98, 26.1)* |
| **Knowledge of modern contraceptive** | | |
| Yes | 1 | 1 |
| No | 1.19(1.09, 1.30)* | 0.88(0.78, 1.00)* |
| **Acceptance toward wife beating** | | |
| Low | 1 | 1 |
| Medium | 1.13(1.06, 1.16)* | 0.96(0.89, 1.02)* |
| High | 1.09(1.02, 1.14)* | 0.89(0.85, 1.00)* |
| **Reading newspaper** | | |
| Not at all | 1 | 1 |
| Less than once a week | 0.69(0.64, 0.74) * | 0.78(0.71, 0.85) * |
| At least once a week | 0.59(0.53, 0.65) * | 0.81(0.72, 0.91)* |

(*Continued*)

**Table 7.** (Continued)

| Variable categories | Unmet need for spacing cRRR (95% CI) | Unmet need for limiting cRRR (95% CI) |
|---|---|---|
| **Listening to a radio** | | |
| Not at all | 1 | 1 |
| Less than once a week | 0.96(0.91, 1.00) | 0.90(0.85, 0.96)* |
| At least once a week | 0.79(0.76, 0.82) * | 0.96(0.90, 1.00)* |
| **Watching TV** | | |
| Not at all | 1 | 1 |
| Less than once a week | 1.10(1.05, 1.16) | 0.91(0.85, 0.98)* |
| At least once a week | 0.91(0.87, 0.95) | 0.88(0.83, 0.93)* |
| **Community media exposure** | | |
| High | 1 | 1 |
| Medium | 1.11(1.04, 1.16) | 1.05(0.98, 1.12) |
| Low | 1.06(0.99, 1.11) | 1.08 (1.01, 1.15) |

**Key:** 1: Reference category; cRRR = Crude Relative Risk Ratio

* significant at p-value <0.20

**Table 8. Multilevel bivariable multinomial logistic regression analysis of factors in resource domain associated with unmet need for contraception in SSA countries, 2016–2021.**

| Variable categories | Unmet need for spacing cRRR (95% CI) | Unmet need for limiting cRRR (95% CI) |
|---|---|---|
| **Wealth index combined** | | |
| Richest | 1 | 1 |
| Richer | 1.17 (1.09, 1.24)* | 1.07(0.99, 1.16) |
| Middle | 1.28(1.21, 1.36)* | 1.13(1.05, 1.21)* |
| Poorer | 1.33(1.25, 1.409)* | 1.10(1.02, 1.18)* |
| Poorest | 1.39(1.31, 1.47)* | 1.09 (1.02, 1.18)* |
| **Community level poverty** | | |
| Low | 1 | 1 |
| Moderate | 1.01(0.96, 1.05) | 1.02(0.96, 1.07) |
| High | 1.03(0.99, 1.08) | 0.96(0.91, 1.02) |
| **Occupational status** | | |
| Employed | 1 | 1 |
| Un employed | 1.37(1.31, 1.43)* | 0.84(0.79, 0.88)* |
| **Distance to a health facility** | | |
| Not a big problem | 1 | 1 |
| Big problem | 1.16(1.12, 1.20)* | 1.13(1.06, 1.16)* |
| **Money needed for treatment** | | |
| Not a big problem | 1 | 1 |
| Big problem | 1.21(1.17, 1.26)* | 1.28(1.23, 1.34)* |
| **Enrolment in a health insurance scheme** | | |
| Yes | | |
| No | 1.54(1.42, 1.65)* | 1.03(0.95, 1.12) |

**Key:** 1: Reference category; cRRR = Crude Relative Risk Ratio

* significant at p-value <0.20

**Table 9. Multilevel bivariable multinomial logistic regression analysis of factors in the decision-making domain associated with unmet need for contraception in SSA countries, 2016–2021.**

| Variable categories | Unmet need for spacing cRRR (95% CI) | Unmet need for limiting cRRR (95% CI) |
|---|---|---|
| **Decision-making for healthcare** | | |
| By husband and others | 1 | 1 |
| Woman and partner | 0.77(0.74, 0.80)* | 0.77(0.73, 0.82)* |
| Woman alone | 0.78(0.74, 0.82)* | 1.52(1.43, 1.62)* |
| **Decision on Purchase** | | |
| By husband and others | 1 | 1 |
| Woman and partner | 0.76(0.73, 0.78)* | 1.18(1.12, 1.23)* |
| Woman alone | 0.95(0.89, 1.01)* | 1.86(1.73, 1.99)* |
| **Decide to visit family** | | |
| By husband and others | 1 | 1 |
| Woman and partner | 0.76(0.73, 0.79)* | 1.10(1.05, 1.16)* |
| Woman alone | 0.86(0.82,0.91)* | 1.47(1.37, 1.56)* |
| **Overall decision making power** | | |
| High | 1 | 1 |
| Medium | 1.15(1.08, 1.21)* | 0.87(0.83, 0.94)* |
| Low | 1.12(1.06, 1.18)* | 0.75(0.71, 0.79)* |
| **Permission to go to a health facility during illness** | | |
| Not a big problem | 1 | 1 |
| Big problem | 1.34(1.28, 1.41) | 1.14(1.08, 1.218) |

**Key:** 1: Reference category; cRRR = Crude Relative Risk Ratio

* significant at p-value <0.20

95% CI: 1.18; 1.53) than women with a higher level of education. Comparing women with a family size of five or more to their counterparts, the odds of unmet need for spacing were 23% (aRRR = 1.23; 95% CI: 1.19, 1.27), and for limiting were 13% (aRRR = 1.19; 95% CI: 1.14, 1.25). Grand multiparous women had an increased likelihood of having unmet needs for spacing and limiting by 4.03 (aRRR = 4.03; 95% CI: 3.09, 5.25) and 6.09 (aRRR = 6.90; 95% CI: 3.52; 13.54) times, respectively. There were 2.74(aRRR = 2.74; 95% CI: 2.62, 2.88) and 3,02 (aRRR = 3.02; 95% CI: 1.45, 6.27) times higher odds of unmet need for limiting among women who are 34 to 49 years old and have five or more children, respectively.

Attitude toward wife beating was also identified as a significant predictor of unmet needs. Women with low acceptance of wife beating had a 19% (aRRR = 0.81; 95% CI: 0.77, 0.96) and 15% (aRRR = 0.85; 95% CI: 0.79, 0.91) lower risk of unmet need for spacing and limiting, respectively. Furthermore, women who listen to the radio at least once a week were 23% (aRRR = 0.77; 95% CI: 0.69, 0.94) and 18% (aRRR = 0.82; 95% CI: 0.78, 0.87) less likely to experience an unmet need for spacing and limiting, respectively. Women in the poorest wealth quintiles had an 18% (aRRR = 1.18; 95% CI: 1.11; 1.25) and 14% (aRRR = 1.14; 95% CI: 1.11, 1.18) higher risk of having unmet needs for spacing and limiting than those in the richest. Unmet need for spacing and limiting was 16% (aRRR = 0.84; 95% CI: 0.80, 0.91) and 13% (aRRR = 0.87; 95% CI: 0.83, 0.94) less likely among women who enrolled in health insurance programs, respectively. Finally, unmet needs for spacing and limiting were 37% (aRRR = 1.37; 95% CI: 1.12, 1.48) and 32% (aRRR = 1.32; 95% CI: 1.11, 1.38) more likely among participants with poor decision-making power, respectively, than in those with a higher power (Table 10).

**Table 10. Results of a multilevel mixed-effect multinomial logistic regression analysis to identify the factors affecting unmet need for contraceptives in SSA, 2016–2021.**

| Variable categories | Model I) | Model II (Influencer domain) | | Model III (resource domain) | | Model-IV (decision making domain) | | Full model | |
|---|---|---|---|---|---|---|---|---|---|
| | | $U_S$ | $U_L$ | $U_S$ | $U_L$ | $U_S$ | $U_L$ | $U_S$ | $U_L$ |
| | | aRRR (95% CI) | aRRR (95% CI) | aRRR (95% CI) | aRRR (95% CI) | aRRR (95% CI) | aRRR (95% CI) | aRRR (95% CI) | aRRR (95% CI) |
| **Current Age** | | | | | | | | | |
| 15–19 | | 1 | 1 | | | | | 1 | 1 |
| 20–34 | | 1.37(1.28, 1.46)** | 0.83 (0.64, 1.00) | | | | | 1.12(0.94, 1.30) | 0.81(0.75, 1.03) |
| 35–49 | | 0.61(0.58, 0.63)* | 2.80(2.67, 2.94)** | | | | | 0.84(0.78, 1.06) | **2.74(2.62, 2.88)**** |
| **Sub-Regions** | | | | | | | | | |
| Southern | | 1 | 1 | | | | | 1 | 1 |
| Central | | 2.93(2.32, 3.71) | 0.96(0.79, 1.16) | | | | | **3.12(2.47, 3.95)**** | **0.**89(0.73, 1.08) |
| Eastern | | 1.94(1.54, 2.42) | 1.19(1.08, 1.32)* | | | | | **2.17(1.73, 2.72)**** | 1.05(0.95, 1.17) |
| Western | | 2.42 (1.82, 3.22) | 0.71(0.60, 0.85)* | | | | | **2.97(2.37, 3.73)**** | 0.90(0.85, 1.03) |
| **Educational status** | | | | | | | | | |
| Higher | | 1 | 1 | | | | | 1 | 1 |
| Secondary | | 1.22(1.12, 1.33)** | 1.23(1.09, 1.39)** | | | | | 1.08(0.99, 1.19) | 1.11(0.98, 1.17) |
| Primary | | 1.29(1.18, 1.41)** | 1.21(1.07, 1.37)** | | | | | **1.11(1.01, 1.21)**** | **1.23(1.09, 1.39)**** |
| No education | | 1.38(1.26, 1.51)** | 1.32 (1.16, 1.49)** | | | | | **1.13(1.03, 1.24)**** | **1.34(1.18, 1.53)**** |
| **Family size** | | | | | | | | | |
| ≤5memeber | | 1 | 1 | | | | | 1 | 1 |
| >5 member | | 1.26(1.22, 1.32)** | 1.09(1.04, 1.14)** | | | | | **1.23(1.19, 1.27)**** | **1.19(1.14, 1.25)**** |
| **Head of HH** | | | | | | | | | |
| Female | | 1 | 1 | | | | | 1 | 1 |
| Male | | 0.66(0.63, 0.68) | 0.65(0.63, 0.70) | | | | | 0.88(0.84 1.11) | 0.71(0.66, 0.75) |
| **Parity** | | | | | | | | | |
| Nulliparous | | 1 | 1 | | | | | 1 | 1 |
| Primiparous | | 2.75(2.16, 3.48) | 1.33(0.70, 2.52)** | | | | | 2.74(2.16, 3.49)** | 1.32(0.69, 2.49) |
| Multiparous | | 3.85(2.98, 4.97) | 4.16(2.13, 8.12)** | | | | | **3.91(3.03, 5.04)**** | **4.09(2.09, 7.98)**** |
| Grand multiparous | | 3.99(3.06, 5.20) | 7.03(3.58, 13.78)** | | | | | **4.03(3.09, 5.25)**** | **6.90(3.52, 13.54**** |
| **Children ever born** | | | | | | | | | |
| No children | | 1 | 1 | | | | | 1 | 1 |
| One | | 0.82(0.64, 1.05)** | 1.49(0.75, 2.98)** | | | | | 0.83(0.65, 1.06) | 1.51(0.76, 3.02) |
| Two to four | | 0.64(0.49, 0.84)** | 1.85(0.89, 3.83)** | | | | | 0.65(0.50, 0.85) | 1.88(0.91, 3.88) |
| Five and more | | 0.61(0.46, 0.79)** | 2.95(1.42, 6.12)** | | | | | 0.92(0.87, 1.12) | **3.02(1.45, 6.27)**** |

*(Continued)*

**Table 10.** (Continued)

| Variable categories | Model I) | Model II (Influencer domain) | | Model III (resource domain) | | Model-IV (decision making domain) | | Full model | |
|---|---|---|---|---|---|---|---|---|---|
| | | $U_S$ | $U_L$ | $U_S$ | $U_L$ | $U_S$ | $U_L$ | $U_S$ | $U_L$ |
| | | aRRR (95% CI) | aRRR (95% CI) | aRRR (95% CI) | aRRR (95% CI) | aRRR (95% CI) | aRRR (95% CI) | aRRR (95% CI) | aRRR (95% CI) |
| Knowledge of modern contraceptive | | | | | | | | | |
| Yes | | 1 | 1 | | | | | 1 | 1 |
| No | | 1.01(0.93, 1.08) | 1.03(0.92, 1.15) | | | | | 0.94(0.87, 1.02) | 1.05(0.94, 1.17) |
| **Overall Acceptance of wife beating** | | | | | | | | | |
| High | | 1 | 1 | | | | | 1 | 1 |
| Medium | | 0.98(0.94, 1.03) | 0.88(.083, 0.93) | | | | | 0.95(0.92, 1.04) | 0.88(0.83, 1.03) |
| Low | | 0.94(0.89, 0.98) | 0.83(0.77, 0.88) | | | | | **0.81(0.77, 0.96)** ** | **0.85(0.79, 0.91)** ** |
| **Reading Newspaper** | | | | | | | | | |
| Not at all | | 1 | 1 | | | | | 1 | 1 |
| Less than once a week | | 0.83(0.78, 0.89) | 1.02(0.94, 1.10) | | | | | 0.95(0.89, 1.02) | 1.02(0.94, 1.11) |
| At least once a week | | 0.79(0.73, 0.87)* | 1.04(0.94, 1.16) | | | | | 0.98(0.94, 1.04) | 1.04(0.94, 1.16) |
| **Listening to radio** | | | | | | | | | |
| Not at all | | 1 | 1 | | | | | 1 | 1 |
| Less than once a week | | 0.95(0.91, 0.98)* | 0.92(0.87, .97) | | | | | 0.97(0.93, 1.01) | 0.95(0.87, 1.07) |
| At least once a week | | 0.87(0.83, 0.90)* | 0.93(0.87, 0.96)* | | | | | 0.87(0.79, 1.04) | 0.92(0.88, 1.07) |
| **Watching Television** | | | | | | | | | |
| Not at all | | 1 | 1 | | | | | 1 | 1 |
| Less than once a week | | 1.02(0.98, 1.07) | 0.99(0.93, 1.06) | | | | | 1.02(0.97, 1.17) | 0.95(0.89, 1.02) |
| At least once a week | | 1.09(1.05, 1.16)* | 1.05(0.99, 1.11)* | | | | | 1.03(0.95, 1.12) | 1.03(0.92, 1.09) |
| **Overall media exposure** | | | | | | | | | |
| Low | | 1 | 1 | | | | | 1 | 1 |
| Medium | | 0.95(0.91, 0.98)* | 0.92(0.87, .97) | | | | | 0.97(0.93, 1.01) | 0.95(0.87, 1.07) |
| High | | 0.87(0.83, 0.90)* | 0.93(0.87, 0.96)* | | | | | **0.77(0.69, 0.94)** ** | **0.82(0.78, 0.87)** ** |
| **Wealth index** | | | | | | | | | |
| Richest | | | | 1 | 1 | | | 1 | 1 |
| Richer | | | | 1.22(1.15, 1.28)** | 1.11(1.04, 1.18)** | | | 1.11(1.00, 1.19) | 0.97(0.94, 1.07) |
| Middle | | | | 1.29(1.22, 1.35)** | 1.15(1.08, 1.22)** | | | 1.07(0.99, 1.24) | 0.88(0.82, 1.05) |
| Poorer | | | | 1.31(1.25, 1.38)** | 1.09(1.02, 1.16)** | | | 1.12(0.98, 1.21) | 0.95(0.89, 1.03) |
| Poorest | | | | 1.33(1.26, 1.39)** | 1.18(1.08, 1.21)** | | | **1.18(1.11, 1.25)** ** | **1.14(1.11, 1.18)** ** |
| **Community level poverty** | | | | | | | | | |
| Low | | | | 1 | 1 | | | 1 | 1 |

(*Continued*)

**Table 10.** (Continued)

| Variable categories | Model I) | Model II (Influencer domain) | | Model III (resource domain) | | Model-IV (decision making domain) | | Full model | |
|---|---|---|---|---|---|---|---|---|---|
| | | U_S | U_L | U_S | U_L | U_S | U_L | U_S | U_L |
| | | aRRR (95% CI) | aRRR (95% CI) | aRRR (95% CI) | aRRR (95% CI) | aRRR (95% CI) | aRRR (95% CI) | aRRR (95% CI) | aRRR (95% CI) |
| Moderate | | | | 0.98(0.94, 1.03) | 0.95(0.90, 1.00) | | | 1.01(0.96, 1.06) | 0.96(.091, 1.02) |
| High | | | | 0.96(0.91, 1.00) | 0.89(0.83, 1.03) | | | 0.99(0.95, 1.05) | 0.95(0.86, 1.04) |
| **Occupational status** | | | | | | | | | |
| Un Employed | | | | 1 | 1 | | | 1 | 1 |
| Employed | | | | 0.96(0.83, 1.08) | 1.23(1.17, 1.28)** | | | 0.87(0.81, 1.06) | 0.96(0.92, 1.02) |
| **Distance to a health facility** | | | | | | | | | |
| Not a big problem | | | | 1 | 1 | | | 1 | 1 |
| Big problem | | | | 0.94(0.96, 1.03) | 0.97(0.93, 1.02) | | | 1.01(0.97, 1.05) | 0.96(.92, 1.11) |
| **Money for treatment** | | | | | | | | | |
| Not a big problem | | | | 1 | 1 | | | 1 | 1 |
| Big problem | | | | 1.15(1.11, 1.19) | 1.27(1.22, 1.33) | | | 1.07(0.97, 1.10) | 1.03(0.98, 1.19) |
| **Health insurance Enrolment** | | | | | | | | | |
| No | | | | 1 | 1 | | | 1 | 1 |
| Yes | | | | 0.70 (0.66, 0.75) | 0.90(0.85, 0.93) | | | **0.84(0.80, 0.91)** ** | **0.87(0.83, 0.94)** ** |
| **Who decides on healthcare** | | | | | | | | | |
| Others | | | | | | 1 | 1 | 1 | 1 |
| Joint decision | | | | | | 1.00(0.93, 1.08) | 1.06(0.9, 1.17) | 0.98(0.93, 1.02) | 1.04(0.93, 1.15) |
| Woman alone | | | | | | 0.86(0.80, 0.94) | 1.13(1.02, 1.24) | 0.93(0.89, 1.03) | 1.06(0.95, 1.18) |
| **Decision on Purchase** | | | | | | | | | |
| Others | | | | | | 1 | 1 | 1 | 1 |
| Joint decision | | | | | | 0.90(0.84, 0.97)** | 1.02(0.92, 1.13) | 1.03(0.96, 1.12) | 0.97(0.87, 1.08) |
| Woman alone | | | | | | 1.16(1.07, 1.26)** | 1.48(1.33, 1.64)** | 1.20(1.11, 1.31) | 1.24(1.11, 1.39)** |
| **Decide to visit family** | | | | | | | | | |
| Others | | | | | | 1 | 1 | 1 | 1 |
| Joint decision | | | | | | 0.95(0.89, 1.00) | 0.93(0.86, 1.00) | 1.02(0.96, 1.07) | 0.91(0.84, 0.99) |
| Woman alone | | | | | | 1.03(0.96, 1.09) | 11(1.02, 1.20)* | 1.02(0.97, 1.04) | 1.05(0.96, 1.14) |
| **Overall decision-making power** | | | | | | | | | |
| High | | | | | | 1 | 1 | 1 | 1 |
| Medium | | | | | | 1.06(0.98, 1.14) | 0.92(0.84, 1.01) | 1.11(1.03, 1.19)** | 0.91(0.82, 1.01) |
| Low | | | | | | 1.17(1.02, 1.35)** | 1.47(1.12, 1.66)** | **1.37(1.12, 1.48)** ** | **1.32(1.11, 1.38)** ** |
| **Permission to go to a health facility** | | | | | | | | | |

*(Continued)*

**Table 10.** (Continued)

| Variable categories | Model I) | Model II (Influencer domain) | | Model III (resource domain) | | Model-IV (decision making domain) | | Full model | |
|---|---|---|---|---|---|---|---|---|---|
| | | U_S | U_L | U_S | U_L | U_S | U_L | U_S | U_L |
| | | aRRR (95% CI) | aRRR (95% CI) | aRRR (95% CI) | aRRR (95% CI) | aRRR (95% CI) | aRRR (95% CI) | aRRR (95% CI) | aRRR (95% CI) |
| Not a big problem | | | | | | 1 | 1 | 1 | 1 |
| Big problem | | | | | | 1.27(1.22, 1.32)** | 1.12(1.07, 1.18) | 1.02(0.97, 1.07) | 1.04(0.99, 1.10) |

**Key:** 1: Reference category; aRRR = Adjusted relative risk ratio

** Statistically significant at p-value <0.05, $U_S$: Unmet need for spacing; $U_L$: unmet need for limiting

## Discussion

This study looked at the level of unmet need for contraception and its correlation with women empowerment indicators across three domains, namely influencer, resource, and decision-making variables, in Sub-Saharan African countries by employing recent DHS data. Accordingly, the pooled prevalence of unmet need for contraception was 26.36% with unmet need for spacing and limiting were 16.74% and 9.62%, respectively. The finding is in tandem with a study conducted in lower and middle-income countries (LMICs) of Asia and Sub-Saharan Africa (24%) [7] and a study conducted in SSA among both married and cohabited women (28.7%) [37]. But it is higher than another study conducted in South and Southeast Asian countries (21%) [38], European countries (17.2%) [39] and lower than some European and Central Asian countries (Turkey, Armenia, Azerbaijan, Serbia, North Macedonia, Bosnia and Herzegovina and Albania) which was approaching 40–50% [40]. The disparities could be due to a complex interaction of socioeconomic (poverty, education, employment), cultural and religious (beliefs, stigma, and misinformation), structural (healthcare infrastructures, access to services, and quality of care), gender-related (power dynamic and family pressure), legal and policy related (legal restrictions and lack of supportive policies), and political(conflict and instability) factors that vary among continents and even countries [7].

As per the results of the analysis, unmet needs for spacing and limiting were associated with educational level, family size, parity, number of children, attitude toward wife beating, media exposure, being in the poorest wealth quintile, enrolment in health insurance schemes, and autonomy in decision-making.

The unmet need for spacing and limiting was higher among women with no formal education. This was supported by a study done in lower and middle-income countries of Asia and Sub-Saharan Africa [7] and primary studies conducted in Indonesia [41], Kenya [42], and Ethiopia [43, 44]. This could be because women no formal education may have deep-rooted misconceptions about contraception, as well as having a limited ability to transform the health education they received about contraception into practice [45]. Women lack a formal education may have inadequate health literacy and access to knowledge on contraceptive method mix, availability, efficacy, and adoption. Furthermore, they may be impacted more by societal norms that restrict open discussions about family planning, making it difficult for them to seek information or services [7, 46]. Because of their lower socioeconomic status and restricted bargaining power within their families and communities, these women may have less agency in making reproductive health decisions. Furthermore, they are less likely to be exposed to FP through the media and other channels, which compromises access to contraceptives and obscures the health. On the other hand, studies supported that, enablement interventions with

advancements in education are in effect to manage women's fertility behavior by satisfying their demand for contraception via providing them with more choices [45, 47]. To deal with the issue of unmet needs, the government, health policymakers, and contraceptive service providers need to concentrate on improving women's educational levels and focusing on non-educated women.

An increase in parity was positively correlated with a higher likelihood of having unmet needs. This finding was in tandem with those of studies carried out in lower and middle-income countries(LMICs) [48], SSA [49], Pakistan [50], and Papua New Guinea [51],. In a similar vein, the odds of having unmet needs were higher among those women with a higher number of children under the age of five than women who do not have any under five children. This finding was in tandem with those of studies conducted in Papua New Guinea [51], and Kenya [42]. One probable explanation is that the more children a woman has, the more likely she is to want to space or limit the number of children she will have [52, 53]. Additionally, this could be due to strong patriarchal mores, a preference for male children over female children, and rural systems that prioritize large families and require women to have many children [51, 54]. And it is apparent that the unmet need is greater in those population groups that are in desperate need of contraception [24, 53]. Last but not least, women who have more children may be preoccupied with caring for them and other home duties, which prevents them from attending health facilities for MNCH services and contraceptives. Thus, contraceptive service providers need to give due emphasis to those women.

Recent studies from different parts of the globe have shown that husbands routinely treat their wives harshly by hitting actively limiting women's access to and autonomy in using family planning methods and other reproductive health services [55–57]. This study also showed an association between a low attitude of women towards a wife beating and less unmet needs for spacing and limiting. This was supported by studies conducted in SSA [55], Myanmar [58], Afghanistan [59], Bangladesh [32], India [60], and Niger [61]. A woman's attitude towards wife-beating is thought to be a proxy for her sense of her status [62]. A woman who views such violence as "unjustifiable" is likely to be conscious of her increased entitlement, self-esteem, and status as well as to reflect favourably on her sense of empowerment [62]. All of these factors contribute to the proper use of contraceptives and decrease unmet needs [63, 64]. This refusal attitude towards wife beating, along with a related stronger sense of entitlement or self-esteem, may improve access to maternal health care during the reproductive age.

Those who had good media exposure were less likely to experience an unmet need for spacing and limiting. This was supported by various studies [50, 65, 66]. Women in South and South-East Asia have gained a better understanding of modern contraceptive methods as a result of proper media use, resulting in a decrease in unmet needs [67, 68]. In addition, women in Pakistan [50], the Philippines [67], India [69], and Mali [70] are more likely to have access to contraceptive uptake if they get adequate media exposure than women who don't. Moreover, in Ethiopia, women without access to media have a greater proportion of unmet needs (38.1%) than women with access to media (25.8%) [71]. The explanation for this could be that learning about various kinds of family planning methods, as well as their usage and side effects, through radio and television can help women to understand contraceptive options, resulting in a decrease in unmet needs [67]. As per a study based on 47 DHS data from sub-Saharan Africa, 44.3% of all family planning-related information is obtained from the media [72]. On the other side, women who receive little media exposure may not have a better grasp of contraception, which cannot result in a positive attitude change towards contraception.

From the resource domains of women empowerment enrolment in HI schemes and being in the poorest wealth quintiles were identified as significant predictors of unmet needs. Women in the poorest wealth quintiles had a higher probability of unmet needs for both

spacing and limiting. This finding was in line with studies conducted in SSA [73], Pakistan [50], Papua New Guinea [51], Nigeria [74], and Ethiopia [44, 75]. This might be because women from low-income families are less likely than women from wealthy families to be able to overcome the financial barriers that prevent them from using contraceptives since they are unable to pay for both the direct and adjunct expenses of doing so. Another aspect would be that when income declines, access to various kinds of information and the affordability of services will be lessened [74].

Women who weren't covered by health insurance were more likely to have unmet needs for spacing and limiting. Various studies showed that being a member of health insurance schemes had a significant effect on the usage of maternal healthcare by 9–11% [76–78]. Large population-based surveys conducted in the United States to investigate the association between prescription contraceptive use and health insurance coverage revealed that insured women were more likely to report using prescription contraceptives than uninsured women [79, 80]. According to a multi-nation study on the uptake of family planning services and health insurance, the unmet need was high in the majority of nations that included family planning services in their health insurance schemes [81]. This is the rationale behind the post-2015 focus on achieving universal health coverage in LMICs and the mounting evidence that health insurance schemes can improve access to maternal healthcare services. As a result, members with health insurance were much more likely to visit health facilities, exposing them to a variety of contraceptive information and reducing unmet needs [82].

External factors relating to women's autonomy can influence contraceptive use [83]. Due to economic dependency, worse negotiating skills, and cultural constraints, many women in SSA nations of patriarchal societies have less control over important domestic decisions [84, 85]. In the current study, having a low decision-making power is related to an unmet need for spacing and limiting. This is backed by studies conducted in Cambodia [86], Bangladesh [27], and Indonesia [87]. This may be because women who have limited decision-making capacity lack independent and shared decision-making on critical issues like financial support to cover some of the direct and indirect expenses for the use of contraceptives [87]. In addition, gender-based power disparities can hinder open discussion between couples about reproductive health decisions and women's access to contraceptive services, which have a negative impact on health outcomes [88]. Thus, governments should work to enhance the surroundings for women's economic and freedom autonomy as this will help to lessen the unmet need for contraception and other reproductive health rights that are not yet fully realised.

The current study offers strengths as well as drawbacks. To begin, this is the first multi-county study to look at the relationship between women empowerment indices and unmet needs, and the findings can help design central-level promotional health policies to mitigate unmet needs. In addition, the most recent nationally representative datasets from 18 SSA countries were used, which were collected using standardized and validated data collection tools and procedures, allowing us to generalize our findings to all married women in the region. Furthermore, due to the clustering effect of DHS data, we used a multilevel multinomial logistic regression approach for better parameter estimates, providing disaggregated information on influencer, resource, and decision-making domains, which is vital for creating contextual interventions. The study has some limitations despite the aforementioned strengths. Due to the cross-sectional nature of the data, it can be difficult to handle recall bias and figure out the causal relationship between the outcome of interest and covariates. Furthermore, given that the community-based surveys attempted to uncover unmet contraceptive needs through a series of interviews, they may have delved into sensitive aspects of participants' contraceptive practices, and thus it is important to acknowledge the possibility of social desirability bias. Finally, because the current study tried to uncover predictors of unmet needs

by using aspects from the three dimensions of women empowerment, there may be other confounders outside of those areas that affect the unmet need that the current study was unable to address.

## Conclusion

Unmet needs for contraception in SSA countries were found to be high. Educational level, family size>5, parity, number of children, attitude toward wife beating, media exposure, being in the poorest wealth quintile, enrolment in health insurance schemes, and autonomy in decision-making were identified as significant determinants of unmet needs for spacing and limiting. Policymakers and contraceptive service providers should place special emphasis on women who lack formal education, are from low-income families, and have large family sizes. Collaboration among governments, civil society, the private sector, and international agencies will be vital in improving media access. Governments should collaborate with insurance providers to increase health insurance coverage alongside incorporating FP procedures within the service package to minimize out-of-pocket costs. NGOs, governments, and program planners should collaborate across sectors to pool resources, advocate for policies, share best practices, and coordinate initiatives to maximize the capacity of women's decision-making autonomy.

## Acknowledgments

We would like to acknowledge the Demographic Health Survey program office for allowing us to access all the relevant DHS data for this study.

## Author Contributions

**Conceptualization:** Aklilu Habte, Biruk Bogale.

**Data curation:** Aklilu Habte.

**Formal analysis:** Aklilu Habte, Aiggan Tamene.

**Investigation:** Aklilu Habte.

**Methodology:** Aklilu Habte, Aiggan Tamene, Biruk Bogale.

**Resources:** Aklilu Habte.

**Software:** Aklilu Habte.

**Supervision:** Aklilu Habte.

**Validation:** Aklilu Habte.

**Writing – original draft:** Aklilu Habte, Aiggan Tamene, Biruk Bogale.

**Writing – review & editing:** Aklilu Habte, Aiggan Tamene, Biruk Bogale.

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
