## [Decision Letter · Decision Letter 0]

9 Aug 2023

PONE-D-23-12294Women empowerment indices and unmet need for contraception among Married women in Sub-Saharan African countries: A multilevel multinomial logistic regression analysis based on Gender Role FrameworkPLOS ONE

Dear Dr. Habte Hailegebireal,

Thank you for submitting your manuscript to PLOS ONE. After careful consideration, we feel that it has merit but does not fully meet PLOS ONE’s publication criteria as it currently stands. Therefore, we invite you to submit a revised version of the manuscript that addresses the points raised during the review process.

Be sure to:Provide clarity on the data extraction process. Give information on whether cohabiting women were included or not; and why. Give explanations on how covariates were handled/controlled for in the course of data analysisImprove on the discussion of the findings of your study to give your readers a better understanding.Suggest practical solutions to the identified problems to cut across individuals, governmental, non governmental and international organizations as a form of recommendation from your research

We look forward to receiving your revised manuscript.

Kind regards,

Olufunmilayo Olufunmilola Banjo, Ph.D

Academic Editor

PLOS ONE

Journal Requirements:

3. Please upload a new copy of Figure 2 as the detail is not clear. Please follow the link for more information: https://blogs.plos.org/plos/2019/06/looking-good-tips-for-creating-your-plos-figures-graphics/" https://blogs.plos.org/plos/2019/06/looking-good-tips-for-creating-your-plos-figures-graphics/

Reviewers' comments:

Reviewer's Responses to Questions

**Comments to the Author**

1. Is the manuscript technically sound, and do the data support the conclusions?

Reviewer #1: Yes

Reviewer #2: Yes

Reviewer #3: Partly

2. Has the statistical analysis been performed appropriately and rigorously? 

Reviewer #1: Yes

Reviewer #2: Yes

Reviewer #3: Yes

3. Have the authors made all data underlying the findings in their manuscript fully available?

Reviewer #1: Yes

Reviewer #2: Yes

Reviewer #3: Yes

4. Is the manuscript presented in an intelligible fashion and written in standard English?

Reviewer #1: Yes

Reviewer #2: Yes

Reviewer #3: Yes

5. Review Comments to the Author

Reviewer #1: In the introduction...

"Promoting access to family planning in SSA regions continues to be a significant barrier to women's reproductive health" [Please kindly rephrase this statement for it not to sound as if the promotion is serving as a barrier]

In your conclusion.

Please kindly provide specific solutions to all the underlying problems that were discovered (not just that government should place emphasis on or be more dedicated).

Give workable, feasible and practicable solutions to the unmet needs.

Aside the government, what role can private and international organization play?

What strategies can be use to make health insurance accessible, affordable or free?

How can FP procedures be subsidized or even free?

In advanced climes what strategies are they employing and how can SSA imbibe it?

Addressing these issues will solidify your conclusion

Thanks

Reviewer #2: General Comment:

The issue of unmet needs in developing countries is significant. This paper is invaluable as foundational material for policy decisions in these nations. The study's success in examining the relationship between female empowerment indicators and large-scale datasets provides insights that can be generalized across developing countries. The following are some specific suggestions for refinement.

Introduction:

1) The paper underscores the importance of unmet contraceptive needs in developing countries and the necessity to explore their association with female empowerment. It might be beneficial to include a discussion on the overall background factors of unmet contraceptive needs from previous research. While the paper touches upon female empowerment as a related factor, a more holistic understanding of other factors seems lacking.

2) Consider clarifying the research question. While the main focus is on the relationship between unmet needs and female empowerment, the charts and results also suggest an emphasis on the prevalence of unmet needs. If the study is a two-tiered approach, indicating the actual state of unmet needs followed by its relationship with female empowerment, it might be better to set two distinct research questions.

Methods:

1) The process of data extraction from the target population for generalization is not entirely clear. Illustrating this process using flowcharts or other visualization tools would enhance clarity. Given the study's aspiration for broad generalization, enhancing transparency in this process would emphasize the research value.

2) The data on unmet needs for contraception were obtained via interviews, which involve privacy-sensitive questions. Potential biases concerning data reliability might arise. Mentioning this potential bias in the limitations section is essential. Additionally, the skill level of the interviewer could influence the classification of data on unmet needs, which also might warrant acknowledgment as a potential bias.

3) There seems to be a lack of control for covariates when examining the relationship between unmet needs and female empowerment. Given the multifaceted nature of female empowerment, it's crucial to explain whether necessary confounding factors have been controlled or to acknowledge it as a limitation.

Results:

1) Chi-square tests are conducted in Tables 4-6. Given the large sample size, even minimal differences might indicate statistical significance. If the main focus here is to showcase participant demographics, highlighting minor differences might impede the overall narrative.

2) The resolution of the figures needs improvement for better clarity and understanding.

Discussion:

1) Highlighting the prevalence of unmet needs is an essential result of this study. The current discussion doesn't address this, and incorporating this aspect might provide a more rounded understanding.

Reviewer #3: Women empowerment indices and unmet need for contraception among Married women in Sub-Saharan African countries: A multilevel multinomial logistic regression analysis based on Gender Role Framework

Abstract

Background

Low women empowerment status, is a known contributing factor to unmet needs for contraception by limiting access to health service delivery points through negative cultural beliefs and practices. Although the unmet need for birth control methods continues to be a major challenge in Sub-Saharan African countries (SSA) countries, little is known about the relationship between unmet needs and domains of women empowerment in the region. Hence, this study aimed at assessing the influence of women empowerment indices on unmet need for contraception in the 18 SSA countries with the most recent DHS report (2016-2021).

Methods

The appended women (IR) file of the most recent standardized Demographic and Health Survey data for SSA nations from 2016–2021 provided the data for this study. A total of 127,650 (with a weighted frequency of 128,939) married women were included and analyzed by using STATA version 16. The Harvard Institute's Gender Roles Framework, which comprised three domains namely, influencing, resource, and decision-making, was employed to identify and categorize the women empowerment indices. The effects of each predictor on the unmet need for spacing and limiting were examined using a multivariable multilevel mixed-effect multinomial logistic regression analysis to consider data clustering. Adjusted relative risk ratio (aRRR) with its corresponding 95% confidence interval was used to declare the statistical significance of the independent variables

Introduction

In developing countries, more than 200 million women who wish to postpone or avoid pregnancy do not use family planning[1]. Rapid population growth in Sub-Saharan African (SSA) region is expected to continue until the end of the twenty-first century[2]. This region contains nearly all of the countries where fertility rates exceed five children per woman and where fertility rates are declining slowly[3, 4]. Promoting access to family planning in SSA regions continues to be a significant barrier to women's reproductive health, according to the Family Planning 2020 (FP2020) report[5]. Meanwhile, the proportion of unmet need for contraception among married women aged 15-49 in SSA exceeds 25% (i.e. about 47 million), compared to 14% in East Asia and 28% in Latin America and the Caribbean[6]. Unmet needs among these women can lead to unwanted pregnancies, unsafe abortions, ill health, and financial hardship for families and society[1]. Access to contraceptives has been noted as one of the ways to help reduce maternal mortality[7, 8]. Despite the government's efforts to make contraceptive services more accessible and affordable, inequalities in contraceptive acceptance rates have been high in SSA, resulting in higher unmet needs [14]. A woman is considered to have an unmet need for FP if she desires to stop or delay/postpone childbearing but does not use any method of contraception[9]. Unmet need for contraception is a significant indicator of the discrepancy between women's intentions for having children and their acceptance of contraception. It also measures how far the nations have come toward achieving universal access to sexual and reproductive health services[10]. Unmet needs for spacing and limiting are the two main categories. An unmet need for spacing is when a woman wants to delay or postpone getting pregnant, whereas an unmet need for limiting is when she wants to have no more children and is not using any form of contraception[9]. This unmet need accounted for the majority (82%) of unintended pregnancies and related unsafe abortions and delayed or no prenatal care [11]. And if all of the demand for contraceptives is met, the number of unintended pregnancies in developing countries, including in the SSA subregion, would drop from 75 million to 22 million annually[9]. With this, there 4 would be 22 million, 15 million, and 90,000 fewer unintended pregnancies, unsafe abortions, and maternal deaths respectively[12, 13]. All these can be achieved by strengthening initiatives that empower women. One dimension that has been shown to influence contraceptive use is women empowerment. Since the launch of the International Conference on Population and Development (ICPD) in Cairo in 1994, women empowerment has become a focus in the population sector[14]. According to the World Bank, Women empowerment is "the process of increasing a woman's or women’s capacity to make meaningful choices and transform them into desired behaviors and results[15]. Others define women empowerment as having the ability to act, independence, involvement, self-direction, consciousness, freedom, movement, and self-confidence [16, 17]. Women's empowerment has become a cornerstone for societal development so it adds value to the creation of more effective, knowledgeable, peaceful, and prosperous societies[18]. Women empowerment has the potential to increase contraceptive use[19, 20]. Low women empowerment status, on the other hand, is a known contributing factor to unmet needs for contraceptives by limiting access to health service delivery points through negative cultural beliefs and practices [21, 22]. In some parts of the SSA, patriarchal rule impairs women's ability to exercise their fundamental reproductive health rights[23]. Furthermore, some family planning initiatives in SSA failed to achieve their intended goal because they failed to take into account the power dynamics between women and their partners[24, 25]. Women in this region are unable to make decisions about whether or not to use contraception, as well as the type of contraception to use[26]. Studies in lower- and middle-income countries have identified an association between unmet needs and poor decision-making abilities in women, low media exposure, a lack of formal education, a high attitude towards wife beating, and unemployment[18, 27, 28]. The Harvard Analytical Framework (gender roles framework) developed by the Harvard Institute for International Development categorizes women empowerment indices into three dimensions: influencer, resource, and decision-making factors[29]. Influencing factors include variables that are symbolic of gender norms and beliefs, such as the gender division of labor, access to, and individual factors [30]. Resource factors include human capital and access to resources, whereas decision-making factors include women's participation in decision-making, including access to and use of resources[29, 30]. 5 Estimates of unmet needs are still a crucial indicator of the success of family planning programs and the amount of demand for contraception in developing nations, but there are some concerns about the precision of these estimates. Although the unmet need for contraceptive methods continues to be a major challenge in SSA countries, little is known about the relationship between unmet needs and domains of women empowerment in the region. So, using a gender analysis framework created by the Harvard Institute, this study looked into the relationship between women's empowerment indicators and unmet contraceptive needs in 18 SSA countries to make useful recommendations

 

Methods Data source, population, and study period A total of 18 demographic and health surveys (DHS) carried out in SSA countries between 2016 and 2021 were considered in this study. The DHS is conducted in 90 countries to collect information on fundamental health indicators. The data for this study came from the appended women's (IR) file, which contains information about contraception, and all important covariates. The study included countries with recent standardized DHS reports from 2016 to 2020 that had complete cases on the relevant variables. Only married, reproductive-age women who were exposed to frequent sexual activity were included. In contrast, infecund women were not supposed to use contraceptives and were excluded from the dataset before analysis [31]. The entire analysis relied on a weighted sample of 128,939 married and fecund women (Table

Why is the focus on married women alone? How about cohabiting, divorced/separated women who are also sexually active?

Why did the authors report significant p-value <0.20?

Why would a confidence interval “0.95(0.89, 1.01)” be significant? See Table 7c

The use of too many explanatory variables make the results difficult to follow

The discussion needs a significant improvement. Being a study that focused on many countries where data were collected in different years, it should be stated that the comparisons made could be misleading

6. PLOS authors have the option to publish the peer review history of their article (what does this mean?). If published, this will include your full peer review and any attached files.

Reviewer #1: **Yes: **Onigbinde Oluwanisola Akanji

Reviewer #2: No

Reviewer #3: No

---

## [Author Response · Author response to Decision Letter 0]

16 Aug 2023

A point-by-point response to editor and reviewers

Authors’ Response to Academic Editor

Dear: Olufunmilayo Olufunmilola Banjo, Ph.D, Academic Editor, Plos One

We thank you for a thorough reading and constructive comments and suggestions on our manuscript and for the opportunity to revise and resubmit. We are pleased to submit the revised version of the manuscript titled “Women empowerment domains and unmet need for contraception among married and cohabiting fecund women in Sub-Saharan Africa: A multilevel analysis based on Gender Role Framework” for your consideration in the special collection of Plos One. On the following pages, you will find our responses to the questions raised by the esteemed reviewer. On behalf of my co-authors, I thank you for your consideration of this resubmission. We appreciate your time and look forward to your response.

Sincerely, 

Aklilu Habte(MPH)(corresponding author)

aklilihabte57@gmail.com

General Comment: 

Thank you for submitting your manuscript to PLOS ONE. After careful consideration, we feel that it has merit but does not fully meet PLOS ONE’s publication criteria as it currently stands. Therefore, we invite you to submit a revised version of the manuscript that addresses the points raised during the review process.

Be sure to:

Comment 1: Provide clarity on the data extraction process. Give information on whether cohabiting women were included or not; and why. Give explanations on how covariates were handled/controlled for in the course of data analysis

Response: The study was based on the data of married and cohabited fecund women and we tried to make more clear on the title of the revised manuscript. For a detailed explanation, we kindly ask you to look at the ‘response to reviewer#3’ section of this document.

Comment 2: Improve the discussion of the findings of your study to give your readers a better understanding.

Response: we entirely accept your and the reviewers' suggestion. Accordingly, we tried to make the discussion section more understandable for the readers by incorporating all the comments of the reviewers. We kindly ask you to look at the ‘discussion section of the revised manuscript’

Comment 3: Suggest practical solutions to the identified problems to cut across individuals, governmental, nongovernmental, and international organizations as a form of recommendation from your research

Response: as per the reviewers' comments and suggestions we made the recommendations more specific and based on the findings of the current study.

Response: Thank you very much for taking the time to review our work and for your positive feedback. We received your thoughtful, and generous review, along with helpful feedback and suggestions, as a valuable contribution to our ongoing work. We have tried to address all the possible comments and suggestions raised by all three reviewers and you in the following session as a response to reviewer#1, reviewer#2, and reviewer#3

Responses on the Journal Requirements:

Comment 1: Please ensure that your manuscript meets PLOS ONE's style requirements, including those for file naming. The PLOS ONE style templates can be found at

Response: I appreciate your suggestion. We put a lot of attention on the journal requirements during the manuscript preparation process from the start and made an effort to follow the guidelines.

Comment 2: In your revised cover letter, please address the following prompts:

2a) If there are ethical or legal restrictions on sharing a de-identified data set, please explain them in detail (e.g., data contain potentially sensitive information, data are owned by a third-party organization, etc.) and who has imposed them (e.g., an ethics committee). Please also provide contact information for a data access committee, ethics committee, or other institutional body to which data requests may be sent.

Response: as we mentioned in the ‘data availability’ section of the initial manuscript, the data supporting the findings of this study can be obtained in anonymized form from the Demographic and Health Survey website at https://www.dhsprogram.com upon reasonable request in the same manner as the authors. The authors did not have any special access privileges that others would not have. So, now, as per your suggestion, we incorporate the statement in the ‘Cover letter’ of the "Revised Manuscript with Track Changes"

2b) If there are no restrictions, please upload the minimal anonymized data set necessary to replicate your study findings as either Supporting Information files or to a stable, public repository and provide us with the relevant URLs, DOIs, or accession numbers.

Response: as per the DHS office rules and regulations, it is impossible to share the data with third parties other than the authors, and the data can be accessed at https://dhsprogram.com/data/dataset_admin/index.cfm based on a reasonable request.

Comment 3: Please upload a new copy of Figure 2 as the detail is not clear.

Response: Thank you for your suggestion, which is helpful to enhance the clarity and resolution of the figure that we already have submitted. As per your suggestion, we have tried to make the figure more clear by changing its DPI to 600

Comment 4: Please review your reference list to ensure that it is complete and correct. If you have cited papers that have been retracted, please include the rationale for doing so in the manuscript text, or remove these references and replace them with relevant current references. Any changes to the reference list should be mentioned in the rebuttal letter that accompanies your revised manuscript. If you need to cite a retracted article, indicate the article’s retracted status in the References list and also include a citation and full reference for the retraction notice.

Response: we have thoroughly reviewed all the references that we have used in the current manuscript and we confidently assure you that we didn’t use any retracted papers.

END________________________________________

 THANK YOU!!!

Authors’ Response To Reviewers(#1, #2 and #3) 

Dear: Reviewer

We thank you for a thorough reading and constructive comments and suggestions on our manuscript and for the opportunity to revise and resubmit. We are pleased to submit the revised version of the manuscript titled “Women empowerment domains and unmet need for contraception among married and cohabiting fecund women in Sub-Saharan Africa: A multilevel analysis based on Gender Role Framework” for your consideration in the special collection of Plos One. On the following pages, you will find our response to your valuable comments and suggestions. On behalf of the co-authors, I thank you for your consideration of this resubmission. We appreciate your time and look forward to our responses to the comments and suggestions raised from your side independently as “Authors’ response to Reviewer#1, #2, and #3 respectively.

Sincerely, 

Aklilu H. (MPH)(corresponding author)

aklilihabte57@gmail.com

Authors’ Response to Reviewer#1

General comments

Comment 1: In the introduction... "Promoting access to family planning in SSA regions continues to be a significant barrier to women's reproductive health" [Please kindly rephrase this statement for it not to sound as if the promotion is serving as a barrier]

Response: We appreciate your thorough assessment and want to apologize for making a remark in our text that might have led readers misled. Now we have corrected the statement and highlighted it in the ‘Introduction’ section of the "Revised Manuscript with Track Changes", Lines 66-68, Page 3

Comment 2: In your conclusion.

Please kindly provide specific solutions to all the underlying problems that were discovered (not just that government should place emphasis on or be more dedicated). Give workable, feasible and practicable solutions to the unmet needs. Aside the government, what role can private and international organization play?

Response: we appreciate your valuable comments and suggestions. Accordingly, we look into our conclusion and we have tried to make it more specific to the findings by addressing other stakeholders. The amended version is highlighted in the ‘Abstract’ section of the "Revised Manuscript with Track Changes", Line 53-60, Page 3

Comment 3: How can FP procedures be subsidized or even free? In advanced climes what strategies are they employing and how can SSA imbibe it?

Response: There are various mechanisms of subsidizing family planning service packages namely government funding, collaborations between governments and private sector entities, health insurance coverage, establishing or supporting clinics that specialize in FP services, and providing subsidies to these clinics can help reduce the cost of FP procedures and other ways that are ratified by the government of each specific country and regions.

Thank you for your constructive comments and suggestions, which we got of them as valuable input in the improvement of our manuscript. We received all of them as a valuable contribution to our ongoing work. 

END_______________________________________

 THANK YOU!!!

Authors' Response to Reviewer#2

General Comments

The issue of unmet needs in developing countries is significant. This paper is invaluable as foundational material for policy decisions in these nations. The study's success in examining the relationship between female empowerment indicators and large-scale datasets provides insights that can be generalized across developing countries. The following are some specific suggestions for refinement.

Response: Thank you very much for taking the time to review our work and for your positive feedback. We received your thoughtful, and generous review, along with helpful feedback and suggestions, as a valuable contribution to our ongoing work. We have tried to address all the possible comments and suggestions raised by you in the following session.

Comment 1: The paper underscores the importance of unmet contraceptive needs in developing countries and the necessity to explore their association with female empowerment. It might be beneficial to include a discussion on the overall background factors of unmet contraceptive needs from previous research. While the paper touches upon female empowerment as a related factor, a more holistic understanding of other factors seems lacking.

Response: thank you for your meticulous review and valuable comment on the ‘introduction’ section of the manuscript. As per your suggestion, we have added some statements that show the importance of exploring the association between women empowerment indicators and the unmet need for contraceptives(Lines 111-119, Page 4) and the background factors that found to influence unmet need(Lines 105-108, Page 4). The added statements were highlighted in the last paragraphs of the ‘Introduction’ section of the "Revised manuscript with track changes"

Comment 2: The data on unmet needs for contraception were obtained via interviews, which involve privacy-sensitive questions. Potential biases concerning data reliability might arise. Mentioning this potential bias in the limitations section is essential. Additionally, the skill level of the interviewer could influence the classification of data on unmet needs, which also might warrant acknowledgment as a potential bias.

Response: we appreciate your suggestion. Of course, there might be a possibility of social desirability bias is inevitable and it must be acknowledged. Accordingly, we added a social desirability bias as one of the limitations in the current study and we highlighted it in the final paragraph of the ‘Discussion’ section of the "Revised manuscript with track changes", Lines 511-514, Page 26

Comment 3: There seems to be a lack of control for covariates when examining the relationship between unmet needs and female empowerment. Given the multifaceted nature of female empowerment, it's crucial to explain whether necessary confounding factors have been controlled or to acknowledge it as a limitation.

Response: We appreciate your insightful inquiry and suggestion. In the current study, we identify the determinants of unmet needs in terms of the indicators. All the variables that were supposed to influence unmet needs were encompassed under one of the three (i.e. either the influencer, resource, or decision-making) domains of a Harvard Analytic framework. We considered each domain as a single level in the multilevel analysis and all the confounders were easy to control by running a multilevel multivariable multinomial logistic regression analysis. but still, we accept that there might be other confounders other than those variables in the women empowerment domains that potentially affect the unmet needs and it’s the right way to acknowledge as a limitation of the current study. Thus, we put and highlighted it in the final paragraph of the ‘Discussion’ section of the "Revised manuscript with track changes", Lines 515-5117, Page 26

 Comment 4: Chi-square tests are conducted in Tables 4-6. Given the large sample size, even minimal differences might indicate statistical significance. If the main focus here is to showcase participant demographics, highlighting minor differences might impede the overall narrative.

Response: We truly value your informed viewpoint, which we also share. This was one of the reasons we didn't choose eligible variables for multivariable regression based solely on the chi-square test and instead performed a bivariable multinomial regression. However, we only used it to demonstrate to the reader the disparity in unmet needs across factors.

Comment 5: The resolution of the figures needs improvement for better clarity and understanding.

Response: thank you for your suggestion. Thank you for your suggestion, which is helpful to enhance the clarity and resolution of the figures that we already have submitted. As per your suggestion, we have tried to make the figures more clear by increasing their DPI from 300 to 600

Comment 6: in the discussion section….Highlighting the prevalence of unmet needs is an essential result of this study. The current discussion doesn't address this, and incorporating this aspect might provide a more rounded understanding.

Response: thank you for your in-depth review and your suggestion. Accordingly, we have added a paragraph that narrates the prevalence of the overall unmet need for contraception and compares and contrasts it with related studies and the possible justifications behind the disparities. We highlighted it in the ‘Discussion’ section of the "Revised manuscript with track changes", Lines 387-399, Page 23

Thank you for your constructive comments and suggestions, which we got of them as valuable input in the improvement of our manuscript. We received all of them as a valuable contribution to our ongoing work. 

THE END____________________________________

 THANK YOU!!!

Authors' Response to Reviewer#3

Comment 1: Why is the focus on married women alone? How about cohabiting, divorced/separated women who are also sexually active?

Response: We appreciate your inquiry. We assure you that the current study was based on married and cohabited fecund women because both of them have almost all comparative demand for contraceptives. That is why we included a variable in Table 3 ‘age at first cohabitation’. To make more clarity we amend the title to ‘among married and cohabited fecund women’. However, we have excluded divorced/widowed women due to some grounds 

1. Studying the unmet need for contraception among different groups of women, such as married and cohabiting women and divorced/widowed, can provide valuable insights into reproductive health and family planning strategies. 

2. Women in cohabitation and marriage women may have different dynamics within their relationships, affecting their decisions about family planning as compared to those who are divorced or widowed. By studying these groups separately, researchers and policymakers can tailor interventions and educational programs to address the unique challenges and preferences faced by each group.

3. Studying unmet needs for contraception within specific contexts allows for a more nuanced understanding of the factors influencing family planning decisions. Factors such as religious beliefs, community norms, and family expectations can differ between these two groups, impacting their access to and use of contraception. On the other hand, studying the determinants of unmet needs among those groups concurrently may lead to biased estimates and to avoid this we preferred to conduct on married women.

4. Analyzing the unmet need for contraception within more homogenous groups can lead to more accurate data interpretation. Studying unmet needs separately allows for more precise targeting of interventions to address specific barriers to contraception use.

5. S compared to divorced women, those in marriage and cohabitation may face different barriers to accessing contraception. For instance, married women might have different concerns related to family planning compared to divorce/widowed women who may be in less formalized relationships. By studying each group separately, interventions can be developed that address their distinct concerns and challenges, leading to more effective outcomes.

We hope you would understand the reason behind our decision behind conducting the study merely on married women who take the lion's share of the population as compared to cohabited one

Comment 2: Why did the authors report a significant p-value <0.20?

Response: thank you for your question. First, it's important to note that most researchers use a p-value arbitrarily of <0.25 or <0.2 to select candidate variables for a multivariable regression. Some scholars and statisticians suggest that in the case of studies with large sample sizes and many covariates (> 20), a p-value of <0.20 is generally considered a higher threshold for statistical significance. we also have existing evidence or prior studies that used a p-value of < 0.2. despite we use of a p-value of 0.2 we also considered the scientific plausibility of variables. We hope you would understand the intention behind we use p-value of <0.20 and we assure you that it didn’t cause any negative effect in our estimations.

Comment 3: Why would a confidence interval “0.95(0.89, 1.01)” be significant? See Table 7c

Response: Thank you for the inquiry. table 7c is the result of a bivariable multinomial regression analysis and the level of significance was determined by a p-value<0.20. how ever it is not significant and that is why we didn’t make it an asterisk on it. 

Comment 4: The use of too many explanatory variables makes the results difficult to follow

Response: we entirely agree with your idea. But as we have relied on the gender analysis framework with variables under all three domains and this resulted in plenty of variables. Haphazardly reducing the variables might lead to biased estimates and jeopardize the functionality of the frameworks this was the reason behind that. 

Comment 7: The discussion needs significant improvement. Being a study that focused on many countries where data were collected in different years, it should be stated that the comparisons made could be misleading

Response: Thank you for your suggestion. We completely agree with you, and we removed the studies and their references from the discussion section because they could lead to inaccurate comparisons.

Comment 8: How generalizable would your results be, having excluded the sample weight of the excluded sample from the computation of the standard errors?

Response: We appreciate your insightful inquiry. As infecund women have no unmet need and most studies conducted on unmet needs excluded them from the analysis to avoid biased estimates. Regarding the generalization, we generalize the finding to all married and cohabited fecund women. For more clarity for the reader, we put the name ‘among married and cohabiting fecund women’ in the tile of the revised manuscript. Regarding the eligibility criteria, we have tried to clarify the statement in the ‘Data source, population, and study period’ section of the "Revised manuscript with track changes", Lines 132-135, Page 5. We hope you would understand our intention behind excluding the infecund women.

Response to minor comments in the review document with track changes

Comment 1: the title is too long and 

Response: we have made a slight change as per your suggestion and now we tried to shorten it. The amended title is highlighted on the ‘title page’ of the "Revised manuscript with track changes", Lines 2-3, Page1,

Comment 2: Presence of redundant ideas in the background section of the abstract

Response: we tried to remove the redundant ideas and the amended version is highlighted in the ‘Abstract’ section of the "Revised manuscript with track changes", Lines 28-32, Page2

Comment 3: regarding a misleading statement written the introduction section

Response: We appreciate your thorough assessment and want to apologize for making a remark in our text that might have misled readers. Now we have corrected the statement and highlighted it in the ‘Introduction’ section of the "Revised Manuscript with Track Changes", Lines 66-68, Page 3

Comment 4: regarding the conclusion section of the abstract

Response: as per your suggestion and other reviewers we made a slight change to the conclusion section to make it more specific and we kindly ask you to look at it.

Comment 5: regarding putting a reference for a statement

Response: we added a reference for that specific statement on Line 87, Page 3 of the "Revised Manuscript with Track Changes"

Thank you for your constructive feedback and suggestions, which are valuable in the improvement of our manuscript!!

THE END____________________________________

 THANK YOU!!!

---

## [Editor Report · Decision Letter 1]

23 Aug 2023

Women empowerment domains and unmet need for contraception among married and cohabiting fecund women in Sub-Saharan Africa: A multilevel analysis based on Gender Role Framework

PONE-D-23-12294R1

Dear Aklilu Habte Hailegebireal,

We’re pleased to inform you that your manuscript has been judged scientifically suitable for publication and will be formally accepted for publication once it meets all outstanding technical requirements.

Kind regards,

Olufunmilayo Olufunmilola Banjo, Ph.D

Academic Editor

PLOS ONE
---

## [Editor Report · Acceptance letter]

29 Aug 2023

PONE-D-23-12294R1 

Women empowerment domains and unmet need for contraception among married and cohabiting fecund women in Sub-Saharan Africa: A multilevel analysis based on Gender Role Framework 

Dear Dr. Habte:

I'm pleased to inform you that your manuscript has been deemed suitable for publication in PLOS ONE. Congratulations! Your manuscript is now with our production department. 

Kind regards, 

on behalf of

Dr. Olufunmilayo Olufunmilola Banjo 

Academic Editor

PLOS ONE